

**A critical review of the use of iron isotopes in atmospheric aerosol research**
Yifan Zhang[1,5,#], Rui Li[2,#], Zachary B. Bunnell[3], Yizhu Chen[1], Guanhong Zhu[4], Jinlong Ma[4],
Guohua Zhang,[1] Tim M. Conway[3,*], Mingjin Tang[1,*]
[1] State Key Laboratory of Advanced Environmental Technology and Guangdong Key
Laboratory of Environmental Protection and Resources Utilization, Guangzhou Institute
of Geochemistry, Chinese Academy of Sciences, Guangzhou, China
[2] Department of Environmental Health, School of Public Health, Shanxi Medical University,
Taiyuan, China
[3] College of Marine Science, University of South Florida, St. Petersburg, Florida, USA
[4] State Key Laboratory of Isotope Geochemistry, CAS Center for Excellence in Deep Earth
Science, Guangzhou Institute of Geochemistry, Chinese Academy of Sciences,
Guangzhou, China
[5] College of Earth and Planetary Sciences, University of Chinese Academy of Sciences, Beijing,
China
[*] Correspondence: Tim Conway (tmconway@usf.edu), Mingjin Tang (mingjintang@gig.ac.cn)
[#] The two authors contributed equivalently to this work.



**Abstract**
Deposition of atmospheric aerosols is recognized as a major source of iron (Fe) to the surface
oceans, where it acts as a key micronutrient for primary productivity and metabolic functions
of marine microbes. Initially, natural desert dust was thought to be the main source of aerosol
Fe, albeit largely insoluble; however, in the last few decades, the role of anthropogenic and
wildfire sources in providing soluble Fe to aerosols has been increasingly recognized. The
stable isotope ratio of Fe ($\delta^{56}$Fe) has emerged as a potential tracer for discriminating and
quantifying sources of aerosol Fe. In this review, we examine the state of the field for using
$\delta^{56}$Fe as an aerosol source tracer, and constraints on endmember signatures. We begin with an
overview of the methodology of $\delta^{56}$Fe analysis for aerosol samples. We then describe
knowledge of $\delta^{56}$Fe endmember signatures of different source materials, and review existing
knowledge of the $\delta^{56}$Fe signature of ambient aerosols collected from around the globe, and how
these measurements can be used to enhance atmospheric Fe deposition modelling. We also
examine the various chemical processing mechanisms which might influence $\delta^{56}$Fe source
signatures of aerosol Fe during its transport in the atmosphere. This review paper is concluded
with a perspective on the state of the field and a call for future work. Overall, we find aerosol
$\delta^{56}$Fe to be a promising tracer, but highlight that greater constraints on both source endmembers
and processing mechanisms are needed to fully utilize this tracer.



## 1 Introduction

Iron (Fe) is the fourth most abundant element in the upper continental crust (UCC), but concentrations of dissolved Fe are very low over much of the surface ocean ($<0.1$ nmol kg$^{-1}$) due to the extremely low solubility of Fe(III) in seawater (Boyd and Ellwood, 2010). Thus, the availability of dissolved Fe may limit marine primary productivity, nitrogen fixation, and ocean-atmosphere carbon exchange over large regions of the surface ocean, especially in High-Nutrient Low Chlorophyll (HNLC) regions (Boyd and Ellwood, 2010; Moore et al., 2013). In early views of the Fe cycle, atmospheric deposition of Fe-bearing aerosols was thought to be the only significant source of dissolved Fe to the surface ocean (Johnson et al., 1997; Tagliabue et al., 2017). However, it is now understood that dissolved Fe is supplied to the oceans by a mixture of external sources that vary in importance by basin, including hydrothermal venting, marine sediment dissolution (both reductive and non-reductive), and atmospheric aerosols (Tagliabue et al., 2017; Conway et al., 2024). Rivers and cryospheric sources may also play a role in some regions. In this view, atmospheric deposition remains a key source of dissolved Fe to the surface ocean, but with great chemical, spatial, and temporal variability that must be understood and characterized.

Atmospheric Fe aerosols reaching the oceans were originally considered to be composed only of natural components, derived through the weathering and erosion of UCC materials; indeed, desert dust, mainly emitted from arid and semi-arid regions, is the dominant source of aerosol Fe on the global scale (Jickells et al., 2005). More recently, however, it has been shown that anthropogenic aerosols can account for a large fraction of soluble aerosol Fe despite of their small contribution to total aerosol Fe (Sholkovitz et al., 2009; Ito et al., 2019; Hamilton





et al., 2020a; Rathod et al., 2020; Ito et al., 2021; Chen et al., 2024), meaning that they can play
an outsized role in influencing marine primary productivity. Anthropogenic aerosols can
include emissions from fossil fuel/biofuel combustion, industry, biomass burning, and
transportation (Sedwick et al., 2007; Sholkovitz et al., 2009; Zhu et al., 2022; Chen et al., 2024).
In this article, non-dust sources are referred to as anthropogenic sources in order to highlight
the impacts of anthropogenic emission. However, we note that wildfire Fe aerosol, composed
of a combination of soil (~64%) and biomass Fe (~36%) (Hamilton et al., 2022; Bunnell et al.,
2025), is a natural source that may be enhanced by anthropogenic land use and climate change.

After emission into the troposphere, natural, anthropogenic, and wildfire aerosols may

undergo various chemical and physical processes, which can 'solubilize' insoluble Fe minerals
to soluble Fe. For example, while Fe in desert dust is 'insoluble' with initial solubility
(operationally defined as the fraction of total Fe that dissolves) being <0.5% (Desboeufs et al.,
2005; Shi et al., 2011; Oakes et al., 2012; Li et al., 2022), several chemical processes can
promote dissolution of insoluble Fe, among which acid processing (i.e. proton-promoted
dissolution) is likely the most important (Meskhidze et al., 2003; Shi et al., 2012; Baker et al.,
2021; Zhang et al., 2023). Indeed, many studies (Baker and Jickells, 2006; Kumar et al., 2010;
Sholkovitz et al., 2012; Mahowald et al., 2018; Sakata et al., 2022) have shown that when
compared to desert dust, Fe solubility can be much higher for ambient aerosol particles
collected over the oceans, suggesting that other sources (e.g., anthropogenic Fe) or atmospheric
processes contribute significantly to soluble aerosol Fe in the troposphere (and for later
deposition to the surface oceans).



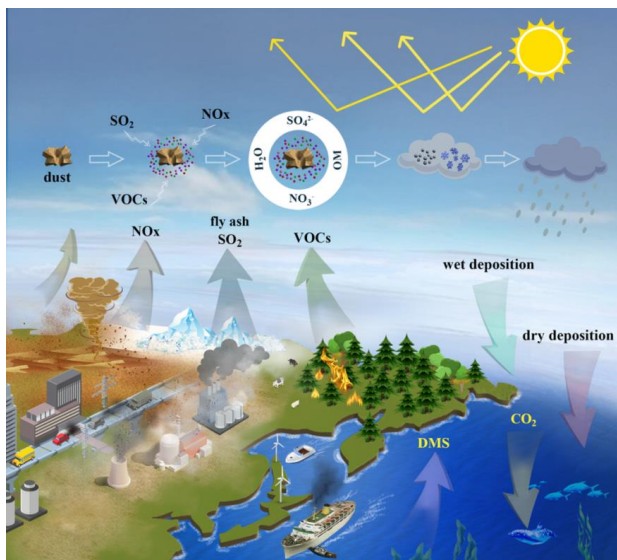


**Figure 1.** Emission, transport, processing and deposition of aerosol Fe. VOCs: volatile organic
compounds; OM: organic materials; DMS: dimethyl sulfide.

Figure 1 depicts emission, transport, processing, and deposition of aerosol Fe. Observed

variability in aerosol Fe flux, composition, and solubility can, in principle, be explained by a
mix of different source emissions and secondary processing in the atmosphere. However, each
type of Fe source exhibits large temporal and spatial variations at the local-global scale, and
thus it remains difficult to quantitatively disentangle their contributions to a sample or location.
Enrichment factors, correlation analysis, and factor analysis (such as positive matrix
factorization) are approaches that have been used for source appointment of total and soluble
aerosol Fe (Chuang et al., 2005; Sedwick et al., 2007; Sholkovitz et al., 2009; Buck et al., 2010;
Desboeufs et al., 2018; Marsay et al., 2022; Zhu et al., 2022; Sakata et al., 2023; Zhang et al.,
2023; Chen et al., 2024; Zhang et al., 2024). However, enrichment factors of other
'anthropogenically-derived' elements (e.g., Cd, Pb, Ni, and Zn) may not always be indicative





101 of anthropogenic Fe (Chester et al., 1993; Shelley et al., 2015), and a consensus has not been

102 reached on either the relative contribution of various sources to soluble aerosol Fe or the factors

103 which control aerosol Fe solubility (Mahowald et al., 2018; Meskhidze et al., 2019; Baker et

104 al., 2021). Single particle analysis can also be very useful for source identification of aerosol

105 Fe in individual particles (Li et al., 2017; Zhu et al., 2020; Liu et al., 2022); nevertheless, it can

106 only examine a limited number of aerosol particles, and thus may not be able to provide

107 quantitative information on the source of Fe for an aerosol sample which typically consists of

108 numerous particles.

109  One tracer that shows recent promise in source quantification of total and soluble Fe

110 within aerosols themselves is the isotopic composition of Fe ($\delta^{56}$Fe). During the last two

111 decades, this parameter has been measured both in marine source materials (Beard et al., 2003a)

112 and later seawater itself (De Jong et al., 2007), providing very useful information for

113 disentangling the marine Fe cycle (Fitzsimmons and Conway, 2023). The developing

114 application of $\delta^{56}$Fe to aerosols has also been shown to be a promising way to differentiate

115 sources of total and soluble Fe in atmospheric aerosols and to quantify their relative importance

116 (Majestic et al., 2009b; Mead et al., 2013; Conway et al., 2019; Kurisu et al., 2021). Wang et

117 al. (2022) briefly summarized recent advances in isotopic compositions of aerosol Fe, and

118 utilized a Bayesian isotopic mixing model (MIXSIAR) to re-evaluate the sources of aerosol Fe

119 based on Fe isotopic data reported in literature. Wei et al. (2024) compiled a global dataset of

120 aerosol Fe isotope measurements, and identified coal combustion as the major anthropogenic

121 source of aerosol Fe by using MIXSIAR to re-analyze the data they compiled.

122  Here, we review the application of Fe isotopic analysis to atmospheric aerosol research in



a comprehensive and critical manner. We summarize the current consensus, underscore existing
discrepancies, and uncertainties/unknowns, as well as outline future research priorities required
to improve the use of Fe isotopic analysis in tracing sources of atmospheric aerosol Fe and
understanding processes that influence its solubility.
**2 Introduction to Fe isotopes and their analysis**
**2.1 Natural Fe isotopes**
Fe has four stable isotopes which occur naturally, namely $^{54}$Fe (5.84%), $^{56}$Fe (91.76%),
$^{57}$Fe (2.12%) and $^{58}$Fe (0.28%). Fe isotopic compositions are typically reported in δ values, as
given in Eq. (1) (Beard and Johnson, 2004; Dauphas and Rouxel, 2006):
$$\delta^{x}Fe(‰)=[\frac{(^{x}Fe/^{54}Fe)_{sample}}{(^{x}Fe/^{54}Fe)_{standard}} - 1]\times1000 \qquad (1)$$

where $x$ is 56, 57 or 58. While some studies report $\delta^{57}$Fe, the convention is to report $\delta^{56}$Fe,
which allows for easy inter-comparability. Here, we use $\delta^{56}$Fe values, although we note that
$\delta^{57}$Fe can be converted to $\delta^{56}$Fe by dividing $\delta^{57}$Fe values by 1.475. Fe isotopic data is also
typically reported relative to the IRMM-014 standard, although some early workers have used
other standards (Beard and Johnson, 2004). In this article we use $\delta^{56}$Fe values relative to
IRMM-014, and for the ease of the reader, have converted published values that were expressed
relative to other standards.
**2.2 Fe isotopic analysis**
Historically, $\delta^{56}$Fe measurements have been made by either multi-collector Thermal
Ionization Mass spectrometry (MC-TIMS) or multi-collector Inductively Coupled Plasma
Mass Spectrometry (MC-ICP-MS). In early works, TIMS was combined with the double-spike
technique, yielding precision of ±0.5‰ (2σ) (Beard and Johnson, 1999; Beard and Johnson,



2004). However, the TIMS-based method, which does not suffer from isobaric interferences,
requires a very long time for each measurement (usually 4-8 hours), and has low Fe ionization
efficiency (Beard et al., 2003b), variable mass bias, and relatively low precision (0.5‰)
compared to natural variability (typically 1-2‰) in open ocean seawater (Dauphas and Rouxel,
2006; Fitzsimmons and Conway, 2023).

The development of MC-ICP-MS, with high ionization efficiency, short analysis time per

measurement, and higher precision (<0.1‰) than TIMS, was crucial to application of $\delta^{56}$Fe to
natural materials (Beard and Johnson, 2004; Zhang et al., 2022). Currently the precision of
$\delta^{56}$Fe measurements via MC-ICP-MS can reach ±0.02 to ±0.05‰ (2σ) (De Jong et al., 2007;
Zhu et al., 2018; Conway et al., 2019), around an order of magnitude better than TIMS;
therefore, MC-ICP-MS has been much more widely used for Fe isotopic analysis.

Drawbacks of MC-ICP-MS are the need of correction for rapid changes in mass bias, and

that measurements can be compromised by isobaric and polyatomic spectral interferences that
must be dealt with using several approaches (Beard et al., 2003b; Beard and Johnson, 2004).
For example, mass-bias must be corrected for using standard-sample bracketing and/or double-
spike techniques, and samples must be cleanly purified from interfering isobars (Cr and Ni)
and matrix elements (e.g., Ca). A further non-trivial challenge of using MC-ICP-MS is the
presence of distinct polyatomic argide interferences (e.g., ArO and ArN) that arise on masses
of $^{54}$Fe, $^{56}$Fe, $^{57}$Fe and $^{58}$Fe from combination of the plasma gas and solvent matrix, and can be
larger than the Fe mass of interest (Weyer and Schwieters, 2003). Furthermore, while elemental
isobars and polyatomic interferences, such as Ni, Cr, and CaO, can be dealt with by purification
and/or correction, argides must be dealt with by changing instrumental conditions or using



‘high’ resolution instruments which can sufficiently resolve Fe peaks from argide interferences
(Weyer and Schwieters, 2003).
The minimum mass of Fe required for isotopic analysis by MC-ICP-MS depends on
several factors, including the procedural blank, analytical uncertainty, and minimum
concentration at which an accurate isotope ratio can be obtained. As MC-ICPMS Neptune Fe
signal intensity decreases, analytical internal standard error (SE) increases relatively
predictably (John and Adkins, 2010; Conway et al., 2013).
Thus, assuming no inaccuracy at lower concentrations, a minimum mass depends on
instrumental signal sensitivity (volts per ng/g) and the minimum acceptable uncertainty for a
sample (e.g., <0.2‰ for Fe). As an example, at the University of South Florida (USF), with a
typical sensitivity of 0.1-0.15 V on $^{56}$Fe per ng of Fe, we are typically able to achieve acceptable
2SE (<0.15‰) in solutions at concentrations of 10 ng/g, a concentration that is >20-40 times
of our chemistry procedural blank (Sieber et al., 2021). This concentration can be achieved by
dissolving 5 ng Fe in 0.5 mL solution. Furthermore, we found that at concentrations below 1V
on $^{56}$Fe, inaccuracies in our measurement of the zero standard begin to occur. In both cases, 5
ng Fe can therefore be considered an absolute minimum mass for analysis. However, 20-100
ng Fe provides isotopic ratios with smaller uncertainty (<0.1‰), approaching the analytical
precision at USF (0.06‰). Similarly, Kurisu et al. (2024) used a minimum of 25 ng Fe to obtain
Fe isotopic data from aerosol samples with suitable uncertainties, but they preferred ~100 ng
where possible. Such precision is often essential for resolving variability between aerosol
samples.
A further consideration for aerosol samples that often include significant filter or





digestion blanks is the blank isotopic composition should be established (and subtracted,
weighed by concentration, from samples) and/or the effect of blanks minimized by having large
sample to blank ratios.

**3 Application of Fe isotopic analysis in aerosol research**

In this section we first present a few examples to illustrate how Fe isotopic analysis may
help constrain sources of dissolved Fe to the ocean (Section 3.1). After that, we review isotopic
compositions of aerosol Fe from relevant sources (Section 3.2), Fe isotopic composition of
ambient aerosols (Section 3.3), modeling studies of isotopic compositions of ambient aerosol
Fe (Section 3.4), and Fe isotopic fractionation induced by chemical processing (Section 3.5).

**3.1 Application of Fe isotopic analysis to marine source attribution**

Since the first successful application of $\delta^{56}$Fe to seawater samples in 2007-2010 (De Jong
et al., 2007; Lacan et al., 2008; John and Adkins, 2010), Fe isotopic analysis has been
increasingly deployed in investigating marine biogeochemistry over the last decade,
significantly increasing our knowledge of sources and cycles of Fe in the ocean (Conway et al.,
2021; Fitzsimmons and Conway, 2023), with a precision of +0.05 to +0.07‰. As this article is
focused on atmospheric aerosols, we do not intend to provide a comprehensive review of the
application of Fe isotopes to marine biogeochemistry; instead, we present a few examples of
relevant ocean observational and modeling studies to illustrate how $\delta^{56}$Fe may help constrain
sources of dissolved Fe to the ocean, focusing on aerosol deposition case studies. For a more
comprehensive discussion of the advances in using $\delta^{56}$Fe to interrogate the marine Fe cycle in
recent years, we instead point the reader to Fitzsimmons and Conway (2023).



Although a few marine $\delta^{56}$Fe water column profiles had been measured by 2012 (Lacan
et al., 2010; Radic et al., 2011; John and Adkins, 2012), the first ocean 'section' of $\delta^{56}$Fe came
in 2014 when Conway and John (2014) reported dissolved Fe and $\delta^{56}$Fe in 510 seawater
samples at 17 stations along the U. S. GEOTRACES GA03 section of the subtropical North
Atlantic Ocean from Woods Hole to Mauritania (Figure 2). A range of $\delta^{56}$Fe values, spanning
from -1.35‰ to +0.8‰, were observed at different regions along this section, reflecting
different sources of dissolved Fe. Most strikingly, much of this transect was characterized by
$\delta^{56}$Fe >+0.1‰, heavier than known sources of Fe to the ocean at the time and attributed to the
net dissolution of atmospheric dust (John and Adkins, 2012; Conway and John, 2014). Further,
by assigning endmember compositions to each identified Fe source (dust, hydrothermal venting,
and sediments), two component mixing was used to quantitatively constrain sources across the
basin (Conway and John, 2014): Saharan dust aerosol was found to be the dominant source for
dissolved Fe along this section (71-87%), with lesser contribution from non-reductive and
reductive sedimentary dissolution (10-19% and 1-4%), and hydrothermal venting (2-6%).
However, that study was not able to use $\delta^{56}$Fe as a source constraint at the very surface, due to
potential influence of biological uptake, and/or anthropogenic aerosols or sediment sources
(Conway and John, 2014; Conway et al., 2019).

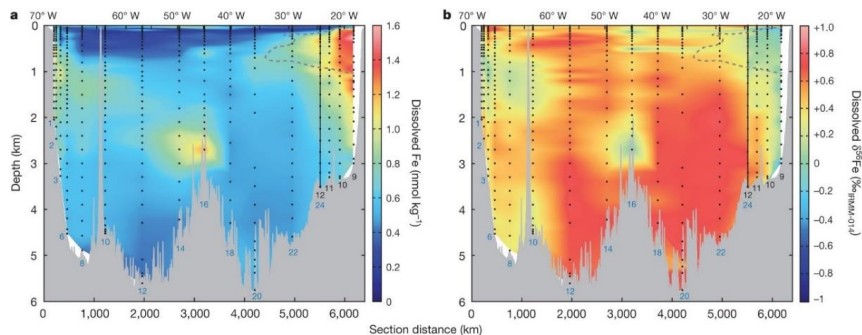




**Figure 2.** Concentrations (a) and $\delta^{56}$Fe values (b) of dissolved Fe in seawater at different depth
along a section of the North Atlantic Ocean from Mauritania and Woods Hole. Station numbers
are shown in blue or black. Reproduced with permission from Conway and John (2014).

A second example of where $\delta^{56}$Fe has been useful for informing our understanding of

atmospheric Fe deposition comes from the North Pacific, where Pinedo-González et al. (2020)
analyzed surface seawater samples collected along a latitudinal section at 158ºW (from 25 to
42ºN) in May 2017. Surface dissolved Fe concentrations peaked at ~35ºN and were lower at
southern and northern ends of this transect, and a similar latitudinal pattern was observed for
dissolved Pb, attributed to deposition of anthropogenic aerosol of both elements. Moreover, the
lowest $\delta^{56}$Fe values (from -0.65‰ to -0.23‰) were observed coincident with the highest
concentrations of dissolved Fe (Pinedo-González et al., 2020), consistent with studies that
showed anthropogenic aerosols to be isotopically light (Kurisu et al., 2016a; Conway et al.,
2019). Using their $\delta^{56}$Fe data and estimates of anthropogenic endmember composition, Pinedo-
González et al. (2020) suggested anthropogenic aerosol to be a significant source (21-59%) of
dissolved Fe in surface seawater between 35 and 40 ºN, at least during the high dust season.
However, other studies of the North Pacific have shown that the primary $\delta^{56}$Fe signature of
aerosol deposition is likely to be strongly attenuated by in situ processing and biological uptake
(Kurisu et al., 2024).

Measurements of surface $\delta^{56}$Fe may therefore be useful in investigating atmospheric

addition of dissolved Fe, but further work is needed to understand the effect of multiple
biogeochemical processes which occur upon dissolution in the ocean. Global biogeochemical




modeling may prove useful here, with recent work by König et al. (2021) incorporating Fe
isotopes into a global model. This study (König et al., 2021) and a follow-up study focused on
the North Pacific (König et al., 2022) found that both external source signatures and
fractionation during Fe complexation by organic ligands and uptake by phytoplankton were
needed to reproduce oceanic deposition patterns, and that the dissolved $\delta^{56}$Fe in surface waters
did not simply follow the footprint of atmospheric deposition.
**3.2 Fe isotopic compositions of aerosols from relevant sources**
In this section we review Fe isotopic compositions of desert dust (Section 3.2.1), raw
materials relevant for anthropogenic emission and biomass burning (Section 3.2.2), and
anthropogenic and biomass burning aerosols (Section 3.2.3). In addition, Table S1 summarizes
key results reported by previous studies.
**3.2.1 Desert dust**
Here we discuss Fe isotopic compositions of desert dust reported by previous work.
Instead of providing a complete literature survey, we review some representative studies. Fe
isotopic compositions appear to be rather homogeneous for UCC, and $\delta^{56}$Fe values fall into a
narrow range centered at around +0.09±0.10‰ (Beard et al., 2003b; Beard and Johnson, 2004;
Poitrasson, 2006).
The average $\delta^{56}$Fe values were determined to be +0.13±0.06‰ and +0.12±0.07‰ for loess
($n$ = 10) and soil samples ($n$ = 10) collected from various locations around the world,
respectively (Beard et al., 2003a). Waeles et al. (2007) measured Fe isotopic compositions of
Chinese and Australian desert dust samples, and the average $\delta^{56}$Fe were determined to be
+0.08±0.04‰ ($n$ = 8) for total Fe and -0.06‰ ($n$ = 2) for soluble Fe, suggesting that the isotopic





composition of soluble Fe in acetate buffer is similar to total Fe. Majestic et al. (2009b)
measured Fe isotopic compositions of road dust and agricultural soil in Phoenix (Arizona,
USA), and $\delta^{56}$Fe values were found to range from -0.10‰ to +0.14‰ for road dust and from -
0.07‰ to 0.04‰ for agricultural soil, both similar to desert dust. In another study (Mead et al.,
2013), average $\delta^{56}$Fe was reported to be 0.09±0.04‰ for three dust samples, namely ATD,
Saharan dust and San Joaquin soil (SRM1709). The $\delta^{56}$Fe values were determined to be -
0.04±0.10‰, -0.05±0.06‰ and 0.21 ± 0.05‰ for ATD, China loess and Xinjiang dust (Li et
al., 2022), with the average being 0.04±0.15‰.

The $\delta^{56}$Fe values were found to range from +0.06‰ to 0.12‰ for loss and paleosol

samples ($n = 32$) from the China Loess Plateau, with the average being 0.09±0.03‰ (Gong et
al., 2017). Chen et al. (2020) analyzed isotopic compositions of HCl-leachable Fe in desert dust
samples ($n = 10$) from different deserts in northern China: one sample from northeastern China
exhibited low $\delta^{56}$Fe (-0.23‰), while $\delta^{56}$Fe showed very small variation and had an average
value of -0.09±0.07‰ for the other nine samples.

One may conclude from previous work that Fe isotopic compositions of desert dust are

very similar to UCC, and most of $\delta^{56}$Fe values fall into a small range (-0.1 to +0.19‰) with an
average value of around +0.09‰. Furthermore, soluble Fe from desert dust appears to be
isotopically similar to total Fe (Waeles et al., 2007; Chen et al., 2020). We note from a recent
synthesis that isotopic compositions of Fe in soil can range from -0.2 to +0.95‰ (Johnson et
al., 2020), showing more variability than in dust, and reflecting the effects of organic matter,
mineralogy, and chemistry.
**3.2.2 Raw materials relevant for anthropogenic emission and biomass burning**





Before we discuss isotopic compositions of aerosol Fe from anthropogenic emission and
biomass burning, it is helpful to review Fe isotopic compositions of relevant raw materials,
including iron ore, fossil fuel, biomass, and so on.
Iron deposits have been found to display very large variations in Fe isotopic compositions
(Johnson et al., 2003; Lobato et al., 2023), with $\delta^{56}$Fe spanning from below -1.5 to around
+2.0‰. The $\delta^{56}$Fe value was determined to be +0.28±0.13‰ for one commercial gasoline
sample in Japan (Kurisu et al., 2016b). To our knowledge, Fe isotopic compositions have not
been reported for other important fossil fuels.
Guelke and Von Blanckenburg (2007) measured isotopic compositions of Fe in higher
plants and plant-available Fe in soils where these plants were grown. Compared to plant-
available Fe in soils, Fe in strategy I plants were isotopically lighter by up to -1.6‰ (Guelke
and Von Blanckenburg, 2007), and younger parts in these plants were more depleted in heavy
Fe. Fe in strategy II plants was isotopically heavier by about +0.2‰ than plant-available Fe in
soils (Guelke and Von Blanckenburg, 2007), and all the parts in these plants had nearly the
same isotopic composition.
In a later study (Kurisu and Takahashi, 2019), average $\delta^{56}$Fe were reported to be
+0.08±0.10‰ for reed and +0.09±0.03‰ for reed burning residual ash, both similar to that for
nearby soil (+0.04±0.20‰). As a result, Kurisu and Takahashi (2019) suggested that biomass
burning could not explain isotopically lighter Fe observed for ambient aerosols.
Kubik et al. (2021) surveyed Fe isotopic compositions of biological samples and found
that in general Fe in strategy I plants is isotopically lighter than strategy II plants. For example,
$\delta^{56}$Fe in stems ranged from -0.72 to -0.08‰ for strategy I plants, and from -1.28‰ to +0.25‰



for strategy II plants; $\delta^{56}$Fe in leaves ranged from -1.04 to +0.23‰ for strategy I plants, and
from -1.20 to 0.24‰ for strategy II plants; $\delta^{56}$Fe in seeds ranged from -1.52 to +0.15‰ for
strategy I plants, and from -1.00 to +0.16‰ for strategy II plants. While various organic
materials have been analyzed for $\delta^{56}$Fe (Guelke and Von Blanckenburg, 2007; Guelke-Stelling
and Von Blanckenburg, 2012; Kurisu and Takahashi, 2019; Kubik et al., 2021), the isotopic
signature for biomass burning as a source of ambient aerosol Fe remains unconstrained.
Additionally, recent work (Hamilton et al., 2022) showed that soil Fe, rather than biomass Fe,
dominates aerosols produced in large scale biomass burning events (wildfires).

In summary, $\delta^{56}$Fe varied greatly from below -1.5‰ to around +2.0‰ for iron ore; no

conclusions can be drawn on $\delta^{56}$Fe for fossil fuel, as observational data are very rare; compared
to desert dust, $\delta^{56}$Fe values can be much lower for plants (and especially strategy I plants).
However, it should be pointed out that $\delta^{56}$Fe values may not be identical for aerosol particles
emitted and their relevant raw materials because Fe isotopic fractionation may occur during
combustion and industrial processes (e.g., iron smelting).
**3.2.3 Anthropogenic aerosols**

Average $\delta^{56}$Fe were determined by Beard et al. (2003a) to be 0.00±0.03‰ ($n$ = 2) for

urban dust (NIST 1649a). Mead et al. (2013) reported the average $\delta^{56}$Fe to be +0.35±0.23‰ ($n$
= 3) for coal fly ash and +0.30±0.17‰ ($n$ = 4) for oil fly ash; in addition, $\delta^{56}$Fe values were
measured to be -0.03±0.13‰ for one diesel particulate matter sample (NIST 1650b) and
0.01±0.12‰ for one urban dust sample (NIST 1649a) (Mead et al., 2013). In another study (Li
et al., 2022), $\delta^{56}$Fe were found to be in the range of +0.05±0.08‰ to +0.75±0.01‰ for three



coal fly ash samples (average: +0.33±0.37‰) and +0.10±0.08‰ for one municipal waste fly
ash (BCR-176R).

In one of the largest steelworks in Europe, $\delta^{56}$Fe values displayed larger variations for

enriched iron ores, ranging from -0.16±0.07 to +1.19±0.14‰ (Flament et al., 2008); for
comparison, $\delta^{56}$Fe values were found to be in the range of +0.53±0.14 to +0.80±0.06‰ for
sintering fly ash and +0.08±0.24‰ for steelwork fly ash. Flament et al. (2008) further
suggested that $\delta^{56}$Fe values reported for sintering and steelwork fly ash fell into the range of
their raw materials (enriched iron ores), implying no significant Fe isotopic fractionation during
steel processes; however, as both enriched iron ores and fly ash displayed large variation in
$\delta^{56}$Fe, this conclusion could be uncertain.

In a parking garage, the average $\delta^{56}$Fe was reported to be +0.15±0.03‰ for coarse

particles (>2.5 μm) (Majestic et al., 2009a), similar to that (+0.18±0.03‰) for fine particles
(<2.5 μm) but slightly larger than that for desert dust. Furthermore, $\delta^{56}$Fe were found to be
+0.19‰ for metallic brake pads, and ranged from +0.42 to +0.61‰ for ceramic brake pads,
from -0.08 to +0.12‰ for tire thread, and from +0.04 to +0.11‰ for waste oil (Majestic et al.,
2009a).

The aforementioned studies measured isotopic compositions of total Fe in urban dust

(Beard et al., 2003a; Mead et al., 2013), coal fly ash (Mead et al., 2013; Li et al., 2022), oil fly
ash (Mead et al., 2013), diesel particulate matter (Mead et al., 2013), municipal waste fly ash
(Li et al., 2022), sintering and steelwork fly ash (Flament et al., 2008), brake pads, tire thread
and waste oil of vehicles (Majestic et al., 2009a), and aerosol particles collected in a parking
garage (Majestic et al., 2009a). The reported $\delta^{56}$Fe values, which show large variations, are



similar to or larger than that for desert dust, and isotopically lighter Fe has hardly been found
in these studies (Beard et al., 2003a; Flament et al., 2008; Majestic et al., 2009a; Mead et al.,
2013; Li et al., 2022).
In contrast, Kurisu et al. (2016b) observed much lower $\delta^{56}$Fe for anthropogenic emissions,
as discussed below. For total Fe in aerosol particles collected in a tunnel in Japan, coarse
particles (>1 μm) had $\delta^{56}$Fe close to UCC while fine particles (<1 μm) exhibited much lower
$\delta^{56}$Fe (as low as -3.2‰) (Kurisu et al., 2016b). Furthermore, $\delta^{56}$Fe were measured to be -
0.08±0.09‰, -0.10±0.03‰, and -0.66±0.09‰ for total Fe in bottom and fly ash of an
incinerator in Japan and total suspended particles (TSP) collected close to its chimney, and -
0.34±0.14‰, -1.97±0.18‰, and -1.25±0.10‰ for soluble Fe (leached using 1 mol/L HCl),
respectively (Kurisu et al., 2016b). Compared to bottom and fly ash, total Fe in TSP exhibited
much lower $\delta^{56}$Fe values; in addition, soluble Fe was isotopically lighter than total Fe. The
following mechanism was proposed to explain their observation (Kurisu et al., 2016b): the
evaporation-condensation process during combustion produced isotopically lighter but more
soluble Fe which was enriched in fly ash and specially in emitted aerosol particles, while most
Fe in bottom ash did not undergo evaporation-condensation. This argument may explain the
discrepancies between the work by Kurisu et al. (2016b) and other studies (Beard et al., 2003a;
Flament et al., 2008; Majestic et al., 2009a; Mead et al., 2013; Li et al., 2022), and can also
explain isotopic compositions of total and soluble Fe in ambient aerosols reported by field
studies (Majestic et al., 2009b; Mead et al., 2013; Kurisu et al., 2016a; Conway et al., 2019;
Kurisu et al., 2019; Kurisu et al., 2021; Zuo et al., 2022).
**3.2.4 Discussion**



381 Since the work by Beard et al. (2003a), there have been a small number of studies which

382 measured isotopic compositions of aerosol Fe from relevant sources. As discussed in Section

383 3.2.1, $\delta^{56}$Fe endmember values have been reasonably well understood for total Fe in desert dust,

384 being similar to UCC. On the other hand, previous studies which measured $\delta^{56}$Fe endmember

385 values for anthropogenic aerosols report a range of isotopic compositions (Beard et al., 2003a;

386 Flament et al., 2008; Majestic et al., 2009a; Mead et al., 2013; Kurisu et al., 2016b; Kurisu and

387 Takahashi, 2019; Li et al., 2022), and the endmember values remain poorly constrained for

388 wildfire aerosols (Bunnell et al., 2025). In general, the number of relevant studies of

389 endmembers is very small, and the number of samples covered by each of these studies is also

390 limited; furthermore, compared to total Fe, $\delta^{56}$Fe endmember values of soluble Fe have been

391 much less examined.

392 The lack of reliable $\delta^{56}$Fe endmember values for total and soluble Fe in non-desert-dust

393 aerosols does limit the use of $\delta^{56}$Fe in tracing and constraining the sources of total and soluble

394 Fe in ambient aerosols. However, we note that the existing studies show much promise.

395 Therefore, we strongly recommend additional measurements of $\delta^{56}$Fe endmember values for

396 total and soluble Fe in non-desert-dust aerosols (aerosols emitted from fossil fuel combustion,

397 biomass burning and metal smelting, for example). In addition, it is very important to continue

398 to explore the dependence of $\delta^{56}$Fe endmember values on particle size, as some previous studies

399 suggested that Fe was isotopically lighter in fine particles when compared to coarse particles

400 (Mead et al., 2013; Kurisu et al., 2016b; Kurisu et al., 2019; Kurisu et al., 2024; Bunnell et al.,

401 2025).

402 **3.3 Fe isotopic composition of ambient aerosols**



Table S2 summarizes previous studies which measured Fe isotopic compositions of

ambient aerosols. Studies published before 2016 and since 2016 are reviewed in Sections 3.3.1

and 3.3.2, respectively. Large variations in $\delta^{56}$Fe have been reported for ambient aerosols: $\delta^{56}$Fe

ranged from -3.53 to +0.48‰ for total aerosol Fe, and from -4.46 to +0.47‰ for soluble aerosol

Fe; by comparison, only a small range of $\delta^{56}$Fe (-0.1 to +0.2‰) has been reported for desert

dust, with an average value of +0.1‰.

**3.3.1 Studies published prior to 2016**

Beard et al. (2003a) reported the first measurement of isotopic signatures of ambient

aerosol Fe. The average $\delta^{56}$Fe of total Fe were determined to be -0.04±0.04‰ for the two TSP

samples collected close to the Gobi desert and +0.11±0.07‰ for the 12 TSP samples collected

at a site in the northwest Pacific (Beard et al., 2003a), both similar to desert dust.

Waeles et al. (2007) measured Fe isotopic compositions of aerosols collected over the

Atlantic and in Barbados, and the average $\delta^{56}$Fe was determined to be +0.04±0.09‰ for total

aerosol Fe and +0.13±0.18‰ for soluble aerosol Fe. No significant difference in isotopic

composition was observed between total and soluble aerosol Fe (Waeles et al., 2007), although

desert dust aerosol could be greatly aged at Barbados which is ~4000 km away from the dust

region.

At an urban site severely affected by a large steel metallurgy plant in France, total aerosol

Fe was found to have an average $\delta^{56}$Fe of +0.14±0.11‰ (Flament et al., 2008), similar to desert

dust. For three aerosol samples collected over the western equatorial Pacific, $\delta^{56}$Fe of total Fe

ranged from +0.27 to +0.38‰ with an average value of +0.33±0.11‰ (Labatut et al., 2014),

slightly higher than desert dust.



Majestic et al. (2009b) investigated isotopic compositions of total Fe in $PM_{2.5}$ and $PM_{10}$
simultaneously collected at a mixed suburban/agricultural site (Phoenix, Arizona, USA). The
first group of $PM_{10}$ samples had $\delta^{56}Fe$ values (centered at around +0.03‰) similar to desert
dust, while the other group of $PM_{10}$ samples exhibited lower $\delta^{56}Fe$ values (centered at -0.18‰),
indicating different Fe sources for the two groups of $PM_{10}$ samples (Majestic et al., 2009b).
Furthermore, the $\delta^{56}Fe$ values of total Fe were substantially lower in $PM_{2.5}$ samples (average:
-0.42±0.14‰) than $PM_{10}$ samples (average: -0.07±0.11‰) (Majestic et al., 2009b), and the
difference was correlated with concentrations (in $PM_{2.5}$) of anthropogenically dominated
elements (such as Pb, V and Cr). Therefore, Majestic et al. (2009b) suggested that
anthropogenic aerosol Fe may explain the lower $\delta^{56}Fe$ for $PM_{2.5}$ than $PM_{10}$.
Mead et al. (2013) collected coarse ($PM_{>2.5}$) and fine ($PM_{2.5}$) aerosol particles in Bermuda
during 2011-2012. As shown in Figure 3, total Fe concentrations were similar for fine and
coarse particles throughout the year, and elevated Fe concentrations in both size fractions were
observed during the high dust season (Mead et al., 2013). For total Fe in coarse particles, $\delta^{56}Fe$
values were similar to the UCC and exhibited no apparent seasonal variation, with an average
value (+0.10±0.06‰, 1σ); in contrast, for total Fe in fine particles, $\delta^{56}Fe$ values were
significantly higher in the high dust season (+0.08±0.05‰, 1σ) than the low dust season (-
0.10±0.14‰, 1σ), implying significant contribution of non-dust sources to aerosol Fe in fine
particles during the low dust season (Mead et al., 2013). Furthermore, biomass burning was
invoked to potentially explain the lighter Fe in fine particles (Mead et al., 2013) due to lower
$\delta^{56}Fe$ reported for various plant matters (Guelke and Von Blanckenburg, 2007).

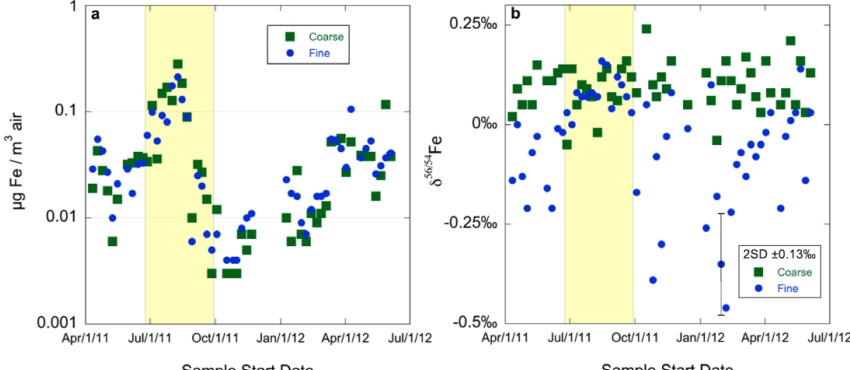


**Figure 3.** Concentrations (a) and $\delta^{56}$Fe values (b) of total Fe in coarse and fine particles

collected in Bermuda in 2011-2012. The shaded area represents the high dust season when the

sampling site was impacted by Saharan dust aerosol. Reproduced with permission from Mead

et al. (2013).

**3.3.2 Studies published since 2016**

For fine (PM$_{2.5}$) and coarse (PM$_{>2.5}$) ambient aerosol particles collected over the

northwestern Pacific, Kurisu et al. (2016a) found that the $\delta^{56}$Fe values of total Fe were

significantly lower in fine particles (-1.17±0.11‰ and -1.72±0.14‰) when compared to coarse

particles (-0.11±0.14‰ and -0.32±0.23‰). Kurisu et al. (2016a) also measured Fe isotopic

compositions for size-fractionated aerosol samples collected at Hiroshima, Japan, and found

$\delta^{56}$Fe to be as low as -2.01‰ for total aerosol Fe and -3.91‰ for soluble aerosol Fe. Figure 4

shows the particle size-dependence of $\delta^{56}$Fe reported by Kurisu et al. (2016a) and highlights

some important features: 1) $\delta^{56}$Fe of total Fe increased with increasing particle size, and at >1.3

µm were similar to desert dust; 2) at a given particle size, $\delta^{56}$Fe were lower for soluble Fe than

total Fe; 3) $\delta^{56}$Fe were lower in August for both total and soluble Fe, when compared to March





(when the impact of Asian desert dust was larger). Furthermore, Fe solubility, which appeared
to be higher in August than March, increased with a decrease in particle size for both months.
Overall, the work by Kurisu et al. (2016a) suggested that anthropogenic Fe, which is more
soluble and isotopically lighter than desert dust Fe, was relatively concentrated in finer particles.
This conclusion was further supported by Fe speciation via X-ray absorption fine structure
(XAFS) spectroscopy analysis (Kurisu et al., 2016a).

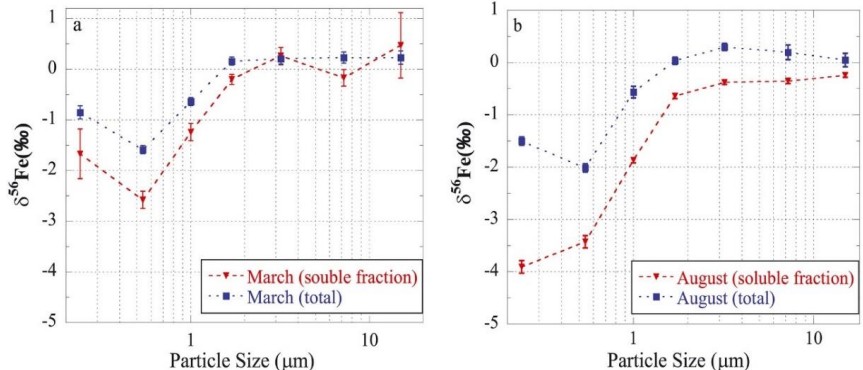


**Figure 4.** Size-resolved $\delta^{56}$Fe of total and soluble Fe for aerosol particles collected at
Hiroshima (Japan) in March (a) and August (b). Reproduced with permission from Kurisu et
al. (2016a).

Kurisu and Takahashi (2019) measured isotopic compositions of aerosol Fe in ambient air
masses affected by a biomass burning event in Tochigi, Japan. Before and after biomass
burning, $\delta^{56}$Fe was measured to be +0.04±0.08‰ on average for coarse particles (>1 μm), while
much lower $\delta^{56}$Fe values were found for fine particles (<1 μm). During the biomass burning
event, $\delta^{56}$Fe values of coarse particles were identical to those before and after the event;
however, $\delta^{56}$Fe values for fine particles, though still lower than those for coarse particles during



the biomass burning event, were found to be higher when compared to those for fine particles
before and after the event. As a result, Kurisu and Takahashi (2019) suggested that the low
$\delta^{56}$Fe measured for fine particles before and after the biomass burning event was due to the
influence of anthropogenic combustion, and that Fe in biomass burning aerosols did not exhibit
low $\delta^{56}$Fe. Kurisu and Takahashi (2019) further speculated that combustion temperature during
the biomass burning event was not high enough to cause Fe isotopic fractionation (and thus
lead to lighter Fe in emitted aerosol particles). In addition, when compared to total Fe, soluble
Fe was found to have lower or similar $\delta^{56}$Fe values (Kurisu and Takahashi, 2019).

Another study by Kurisu et al. (2019) examined Fe solubility and isotopic compositions

of size-resolved aerosol particles collected in Chiba, Japan, heavily impacted by steel plant
emissions, and found higher Fe solubility in fine particles (<1.3 μm) than coarse particles (>1.3
μm). Moreover, $\delta^{56}$Fe ranged from -0.42 to +0.33‰ for coarse particles, likely reflecting Fe
isotopic compositions in steel slags and raw materials; Fe in fine particles appeared to be
isotopically lighter ($\delta^{56}$Fe in the range of -3.53 to -0.37‰), implying Fe fractionation during
evaporation under high temperature (Kurisu et al., 2019). Kurisu et al. (2019) also found that
soluble Fe (extracted using ultrapure water or simulated rainwater) was isotopically lighter than
total Fe.

Conway et al. (2019) measured solubility and isotopic compositions of aerosol Fe

collected over the North Atlantic during winter. Air masses originating from the Saharan region
were characterized by higher concentrations of aerosol Fe and lower Fe solubility, and average
$\delta^{56}$Fe were reported to be +0.12±0.03‰ for total Fe and +0.09±0.02‰ for soluble Fe (Figure
5), similar to desert dust (Conway et al., 2019). In contrast, air masses originating from Europe



and North America exhibited lower concentrations of aerosol Fe but much higher Fe solubility,
and compared to Saharan air masses, $\delta^{56}$Fe in European and North American air masses were
much lower for soluble Fe (mean: -0.91‰) but only slightly lower for total Fe
(mean: -0.12±0.06‰). As a result, Conway et al. (2019) suggested that anthropogenic Fe with
higher solubility but lower $\delta^{56}$Fe made a large contribution to total Fe and especially soluble
Fe in European and North American air masses. This work further utilized a two-component
mixing model with assigned endmember $\delta^{56}$Fe values for natural and anthropogenic Fe (+0.09‰
and -1.60‰, respectively) to constrain the sources of soluble Fe. Fossil fuel combustion
comprised up to ~50-100% of soluble Fe for air masses originating from North America and
Europe (Conway et al., 2019), while the Saharan air masses were characterized by ~100%
natural Fe from desert dust.

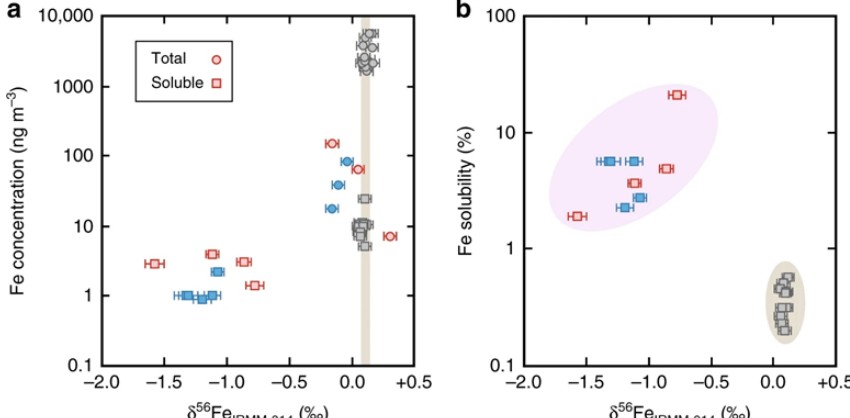


**Figure 5.** Measured $\delta^{56}$Fe of total and soluble Fe versus (a) Fe concentrations and (b) Fe
solubility. Aerosol particles from different air masses are separated with colors (gray: Saharan
air masses; red: European air masses; blue: North American air masses). Reproduced with
permission from Conway et al. (2019).





Kurisu et al. (2021) investigated Fe solubility, speciation, and isotopic compositions of
fine (<2.5 μm) and coarse (>2.5 μm) particles collected over the northwestern Pacific, and the
air masses they sampled broadly came from East Asia, central/eastern Pacific, or from over the
northern Pacific. The average $\delta^{56}$Fe value of total Fe was +0.25±0.14‰ and +0.23±0.17‰ for
coarse and fine particles in air masses from central/eastern Pacific, respectively, and
+0.14±0.10‰ and +0.43±0.17‰ from the North Pacific; from East Asia, average $\delta^{56}$Fe of total
Fe was determined to be -1.10±0.63‰ for fine particles, much lower than that for coarse
particles (-0.02±0.14‰, close to desert dust). For air masses from East Asia, $\delta^{56}$Fe values of
total Fe were found to be negatively correlated with Fe solubility (Kurisu et al., 2021),
suggesting that combustion Fe had lower $\delta^{56}$Fe and higher solubility; the relative contribution
of combustion to total aerosol Fe could be up to 50% for fine particles and 6% for coarse
particles, estimated using obtained Fe isotopic data. For the three aerosol samples which
contained enough soluble Fe for isotopic analysis, $\delta^{56}$Fe were significantly lower for soluble
Fe (as low as -2.23±0.04‰) than total Fe (Kurisu et al., 2021), indicating preferential
dissolution of Fe with lower $\delta^{56}$Fe (for example, perhaps anthropogenic or combustion Fe).
Zuo et al. (2022) analyzed Fe isotopic compositions of magnetic materials in fine (<2.5
μm) and coarse particles (>2.5 μm) collected in Beijing during March-May 2021 when dust
storms occurred. The average $\delta^{56}$Fe was determined to be +0.15±0.04‰ in coarse particles
(similar to desert dust), indicating desert dust as the dominant source for magnetic Fe in coarse
particles. For fine particles, the average $\delta^{56}$Fe was -0.57±0.08‰ during the non-dust-storm
period, suggesting important contribution from anthropogenic sources; it increased



to -0.26±0.21‰ during the dust-storm period, implying an enhanced contribution of desert dust
when compared to the non-dust-storm period.

A recent study (Kurisu et al., 2024) measured isotopic composition of total and soluble Fe

in TSP and size-resolved aerosol particles collected over the subarctic North Pacific. Overall,
for a given sample, $\delta^{56}$Fe was similar or lower for soluble Fe, when compared to total Fe
(Kurisu et al., 2024): $\delta^{56}$Fe ranged from -0.5 to +0.4‰ and from -1.87 to +0.28‰ for total and
soluble Fe in TSP, and from -2.8 to +0.5‰ for total Fe in the size-resolved aerosol samples. A
two-component mixing model was used to estimate the contribution of combustion to total and
soluble aerosol Fe, assuming that crustal and combustion Fe had endmembers values of +0.1‰
and -4.3‰, respectively. The contribution of combustion could reach up to 13% for total Fe
and 45% for soluble Fe in TSP samples, and was highest in the coastal region of East Asia
(Kurisu et al., 2024).

Hsieh and Ho (2024) reported isotopic compositions of total and soluble (ultrapure water

leached) Fe for size-resolved aerosol particles collected on a small islet in the East China Sea.
The $\delta^{56}$Fe values of total Fe ranged from -0.08 to +0.16‰ for large particles (>1.6 μm),
showing no dependence on particle size; however, they decreased to -1.16 to -0.06‰ and
further to -3.35 to -0.45‰ for particles in the size range of 1.0-1.6 and 0.57-1.0 μm,
respectively (Hsieh and Ho, 2024). Fe solubility was significantly correlated with $\delta^{56}$Fe of total
Fe, suggesting significant contribution of anthropogenic sources to soluble Fe. In addition,
soluble Fe was isotopically lighter than total Fe, especially for fine particles (0.57-1.0 and 1.0-
1.6 μm). The contribution of anthropogenic sources was estimated to be 3.5-7.6% for total Fe



and 22-85% for soluble Fe (Hsieh and Ho, 2024), using a two-component mixing model which
assumed $\delta^{56}$Fe endmember values to be +0.1 and -4.4‰ for lithogenic and anthropogenic Fe.

Most recently, Bunnell et al. (2025) measured $\delta^{56}$Fe of TSP as well as coarse (>0.95 μm)

and fine (<0.95 μm) aerosols collected from the North Pacific GEOTRACES GP15 section
(along 152 °W, from 52 °N to 20 °S) during a low dust season (September-November 2018). In
Asian Outflow deployments, total Fe in TSP and coarse particles had $\delta^{56}$Fe values similar to
the UCC, ranging from -0.03 and +0.07‰ and +0.02 to +0.20‰, respectively (Bunnell et al.,
2025); however, total Fe in fine particles within the Asian Outflow was isotopically lighter
(with $\delta^{56}$Fe in the range of -0.39 to -0.25‰), indicative of the contribution of anthropogenic
Fe. In the Equatorial Pacific, $\delta^{56}$Fe of total Fe ranged from +0.15 to +0.41‰ for TSP, +0.15 to
+0.27‰ for coarse particles, and +0.06 to +0.36‰ for fine particles (Bunnell et al., 2025),
similar to or larger than the UCC. Heavier Fe observed in the Equatorial Pacific was attributed
to wildfire soil Fe from western North America (Californian wildfires), based on the
information that up to 64% of aerosol Fe produced in wildfire is derived from soil due to
pyroconvective entrainment (Hamilton et al., 2022) and that $\delta^{56}$Fe are in the range of +0.20 to
+0.95‰ for Californian soils (Johnson et al., 2020). Although large differences in $\delta^{56}$Fe of total
Fe were observed between Asian Outflow and Equatorial Pacific aerosols, soluble aerosol Fe
was generally light ($\delta^{56}$Fe ranging from -1.28 to +0.02‰) throughout the transect (Bunnell et
al., 2025), attributed to influence of anthropogenic Fe.
**3.3.3 Discussion**

Most field measurements found for total Fe in ambient aerosols, the $\delta^{56}$Fe values were

similar to or lower than UCC (Beard et al., 2003a; Waeles et al., 2007; Flament et al., 2008;



Majestic et al., 2009b; Mead et al., 2013; Kurisu et al., 2016a; Conway et al., 2019; Kurisu et
al., 2019; Kurisu and Takahashi, 2019; Kurisu et al., 2021; Zuo et al., 2022; Kurisu et al., 2024).
From these studies which reported lower $\delta^{56}$Fe for total Fe in ambient aerosols (Majestic et al.,
2009b; Mead et al., 2013; Kurisu et al., 2016a; Conway et al., 2019; Kurisu et al., 2019; Kurisu
and Takahashi, 2019; Kurisu et al., 2021; Zuo et al., 2022), two features can generally be
identified. First, when compared to those dominated by desert dust aerosol, $\delta^{56}$Fe values for
total aerosol Fe were lower in air masses severely influenced by anthropogenic emissions
(Majestic et al., 2009b; Mead et al., 2013; Kurisu et al., 2016a; Conway et al., 2019; Kurisu et
al., 2021; Zuo et al., 2022). Second, $\delta^{56}$Fe values frequently appeared to be lower for smaller
particles than larger particles (Majestic et al., 2009b; Mead et al., 2013; Kurisu et al., 2016a;
Kurisu et al., 2019; Kurisu and Takahashi, 2019; Kurisu et al., 2021; Zuo et al., 2022; Hsieh
and Ho, 2024).

Only a few studies (Waeles et al., 2007; Kurisu et al., 2016a; Conway et al., 2019; Kurisu

et al., 2019; Kurisu and Takahashi, 2019; Kurisu et al., 2021; Bunnell et al., 2025) reported
isotopic compositions of soluble Fe in ambient aerosols, and some of them (Kurisu et al., 2016a;
Conway et al., 2019; Kurisu et al., 2019; Kurisu and Takahashi, 2019; Kurisu et al., 2021;
Bunnell et al., 2025) reported lower $\delta^{56}$Fe for soluble aerosol Fe (relative to UCC). Similar to
total Fe, $\delta^{56}$Fe of soluble Fe were found be lower in air masses with severe impacts by
anthropogenic emissions (Kurisu et al., 2016a; Conway et al., 2019), when compared to air
masses dominated by desert dust; furthermore, $\delta^{56}$Fe of soluble Fe decreased with decrease in
particle size (Kurisu et al., 2016a; Kurisu et al., 2019). Compared to total aerosol Fe, soluble
aerosol Fe was found to be isotopically lighter (Kurisu et al., 2016a; Conway et al., 2019;





Kurisu et al., 2019; Kurisu et al., 2021), especially for smaller particles in air masses severely
affected by anthropogenic emissions.

The overall features of isotopic compositions of total and soluble Fe in ambient aerosols,

revealed by a limited number of field studies, can be possibly explained by anthropogenic Fe
(such as fossil fuel combustion, industrial emission, and etc.) which is isotopically lighter, more
soluble, and enriched in fine particles, when compared to desert dust Fe. This explanation is
elaborated upon below. First, as anthropogenic Fe is isotopically lighter than desert dust Fe,
lower $\delta^{56}$Fe are expected in air masses severely affected by anthropogenic emissions, when
compared to air masses dominated by desert dust. Second, since in general anthropogenic Fe
is enriched in fine particles while desert dust Fe is enriched in coarse particles, we expect Fe
in fine particles to be isotopically lighter than coarse particles. Lastly, anthropogenic Fe is more
soluble than desert dust Fe, and therefore the net $\delta^{56}$Fe values are lower for overall soluble Fe
than total Fe.

Some studies (Majestic et al., 2009b; Kurisu et al., 2016a; Conway et al., 2019; Kurisu et

al., 2019; Kurisu et al., 2021; Zuo et al., 2022) have suggested that isotopically lighter $\delta^{56}$Fe in
ambient aerosols can be explained by anthropogenic emission (such as fossil fuel combustion
and metal smelting). Mead et al. (2013) suggested biomass burning as the likely source of
isotopically lighter Fe in ambient aerosols, while more recent studies (Conway et al., 2019;
Kurisu and Takahashi, 2019) did not lend support to this idea. Furthermore, several studies
(Labatut et al., 2014; Conway et al., 2019; Kurisu et al., 2021; Kurisu et al., 2024; Bunnell et
al., 2025) observed significantly higher $\delta^{56}$Fe (than UCC) for ambient aerosol Fe. This
observation has been variably attributed to coal and oil fly ash (Mead et al., 2013; Kurisu et al.,



2016b; Li et al., 2022), or most recently to wildfire emissions (Bunnell et al., 2025). Further
work to investigate the source signatures of different emissions is required (as discussed in
Section 3.2).
**3.4 Modeling studies of isotopic compositions of ambient aerosol Fe**
Conway et al. (2019) compared measured solubility and isotopic compositions of aerosol
Fe over the Atlantic with those predicted using a 3-D model (Community Atmosphere Model
v4, CAM4), in which $\delta^{56}$Fe endmember values were set to +0.09‰ for desert dust and -1.6‰
for anthropogenic aerosol, as constrained by their observation. Compared to measurements,
although the default CAM4 model could reasonably well reproduce Fe solubility for European
and North American aerosols, it overestimated Fe solubility for Saharan aerosols; furthermore,
it failed to simulate $\delta^{56}$Fe for soluble aerosol Fe (Conway et al., 2019). The model could much
better reproduce measured Fe solubility and $\delta^{56}$Fe only if Fe solubility of dust aerosol was
reduced from 25% to 10% and anthropogenic Fe aerosol emission was increased by a factor of
5. Therefore, Conway et al. (2019) suggested that the contribution of anthropogenic emission
to soluble aerosol Fe might have been significantly underestimated.
Kurisu et al. (2021) employed a 3-D model (IMPACT) to simulate isotopic compositions
of total Fe in fine (<2.5 μm) and coarse (>2.5 μm) particles over the northwestern Pacific, and
in their simulation $\delta^{56}$Fe were set to 0‰ for desert dust and -4.7‰ to -3.9‰ for combustion
aerosol. The modelled $\delta^{56}$Fe agreed well with observations for fine particles but were
significantly lower for coarse particles. This underestimation may result from using the same
endmember $\delta^{56}$Fe values for fine and coarse particles in the model (Kurisu et al., 2021); in
other words, if the endmember $\delta^{56}$Fe value of combustion Fe was set to be larger for coarse





particles than fine particles, the model may reproduce the observed $\delta^{56}$Fe for both fine and
coarse particles.
A very recent study (Bunnell et al., 2025) compared their measured total Fe concentrations
and isotopic compositions for aerosols collected from the North Pacific GEOTRACES GP15
section with those simulated by Community Atmosphere Model v6 (CAM6). Based on
previous work by Conway et al. (2019), using the initial endmember values (+0.1‰ for dust,
and -1.60‰ for anthropogenic, wildfire and shipping aerosols) in the model caused large
underestimations in $\delta^{56}$Fe (Bunnell et al., 2025). When compared to the initial values, the
optimal endmember values (which led to best agreement between measured and simulated
$\delta^{56}$Fe of ambient total aerosol Fe) remained unchanged for dust and anthropogenic aerosol, but
increased to +0.8‰ and +0.5‰ for wildfire and shipping, respectively (Bunnell et al., 2025).
Overall, modeling studies which simulate isotopic compositions of ambient aerosol Fe are
very limited at present. As measurements of isotopic compositions of ambient aerosol Fe and
aerosol Fe emitted from various sources are increasing, it is expected that more modeling
studies will include isotopic compositions to better constrain sources of aerosol Fe.
**3.5 Fe isotopic fractionation induced by chemical processing**
Chemical processes which change the speciation of Fe in the environment may also lead
to isotopic fractionation that may attenuate or drive isotopic signatures observed in ambient
aerosols. As summarized in Table 1, in this section we discuss previous studies which
investigated Fe isotopic fractionation induced by a few types of reactions of potential relevance
for atmospheric aerosol Fe, namely proton-promoted dissolution, redox reaction, reductive
dissolution, ligand complexation and ligand-promoted dissolution. We refer the readers to Yin





et al. (2023) for a more comprehensive discussion of Fe isotopic fractionation induced by
biogeochemical processes.

**Table 1.** Fe isotopic fractionation caused by chemical processes relevant for atmospheric
aerosol Fe. In this table, pFe(II) represents particulate Fe, dFe(III) and dFe(II) represent
dissolved Fe(III) and Fe(II), and dFe(III)-ligand represent dissolved Fe(III) complexed with
ligands.

| chemical process | chemical formula | $\Delta^{56}Fe$ | Ref. |
|---|---|---|---|
| proton-promoted dissolution | pFe(III) → dFe(III) | -1.8‰ to +0‰ | a |
| redox reaction | dFe(III) → dFe(II) | -2.8‰ | b |
| reductive dissolution | pFe(III) → dFe(II) | -2.2‰ to -1‰ | c |
| ligand complexation | dFe(III) + ligand → dFe(III)-ligand | +0.3‰ to +0.6‰ | d |
| ligand-promoted dissolution | pFe(III) + ligand → dFe(III)-ligand | -1.3‰ to +0‰ | e |

a: Skulan et al. (2002), Wiederhold et al. (2006), Chapman et al. (2009), Kiczka et al. (2010);
b: Johnson et al. (2002), Welch et al. (2003);
c: Wiederhold et al. (2006);
d: Wiederhold et al. (2006), Dideriksen et al. (2008);
e: Brantley et al. (2001), Wiederhold et al. (2006), Chapman et al. (2009), Kiczka et al. (2010).

**3.5.1 Proton-promoted dissolution**

Wiederhold et al. (2006) examined goethite dissolution in 0.5 mol/L HCl and found that

the isotopic composition for dissolved Fe in the solution was not statistically different from
that of original goethite over the entire experiment (up to 315 days), suggesting that goethite
dissolution in HCl solution did not lead to Fe isotopic fractionation. However, some other





studies (Skulan et al., 2002; Chapman et al., 2009; Kiczka et al., 2010) suggest that proton-
promoted dissolution could lead to significant Fe isotopic fractionation. For example, in the
early stage of hematite dissolution in 0.9 mol/L HCl, dissolved Fe was much isotopically lighter
due to kinetic isotopic fractionation and $\Delta^{56}$Fe reached -1.32±0.12‰ (Skulan et al., 2002);
however, the effect of kinetic isotopic fractionation became smaller with time and eventually
isotopic compositions were identical between dissolved Fe and hematite. Significant Fe
isotopic fractionation, with $\Delta^{56}$Fe being as low as -1.8‰, was reported for dissolution of granite
and basalt in 0.5 mol/L HCl (Chapman et al., 2009). Dissolution of biotite and chlorite in HCl
solution (pH = 4) also led to significant Fe isotopic fractionation (Kiczka et al., 2010), and
$\Delta^{56}$Fe reached as low as -1.4‰; furthermore, Kiczka et al. (2010) found the presence of $K^+$ in
the solution would facilitate isotopic fractionation.

It remains unclear why previous studies (Skulan et al., 2002; Wiederhold et al., 2006;

Chapman et al., 2009; Kiczka et al., 2010) reported different Fe isotopic fractionation effects
induced by proton-promoted dissolution, and difference in Fe mineralogy (thus bonding
strength) may play a role.

**3.5.2 Redox reaction and reductive dissolution**

Johnson et al. (2002) investigated Fe isotopic fractionation for aqueous redox of Fe(III)

to Fe(II) at room temperature, and observed similar degrees of Fe isotopic fractionation at pH
of 2 and 3: Fe(II) was isotopically lighter than Fe(III), and $\Delta^{56}$Fe was determined to
be -2.75±0.15‰ when the reaction reached equilibrium. Welch et al. (2003) examined the
effects of temperatures (0 and 22 ℃) and Cl$^-$ concentrations on Fe isotopic fractionation in
aqueous redox of Fe(III) to Fe(II) and observed significant enrichment of isotopically lighter





Fe in Fe(II). The average $\Delta^{56}$Fe values were -2.76±0.09‰, -2.87±0.22‰, and -2.76±0.06‰ at
22 °C when Cl$^-$ concentrations in the solution were 0, 11, and 111 mmol/L (Welch et al., 2003),
respectively, suggesting no significant impact of Cl$^-$; moreover, a decrease in temperature from
22 to 0 °C resulted in larger Fe isotopic fractionation, and the average $\Delta^{56}$Fe was -3.25±0.38‰
at 0 °C in the absence of Cl$^-$ in the aqueous solutions. The two previous studies (Johnson et al.,
2002; Welch et al., 2003) both suggested that isotopic fractionation occurred when Fe(III) was
reduced to Fe(II) in the solution, with Fe(II) being isotopically lighter than Fe(III).
Wiederhold et al. (2006) explored reductive dissolution of goethite in 0.5 mmol/L oxalate
solution irradiated using a solar simulator. The dissolved fraction was isotopically lighter in the
initial stage, with $\Delta^{56}$Fe being as low as -1.6 ‰ (Wiederhold et al., 2006); this effect gradually
became smaller and the isotopic composition of dissolved Fe eventually was identical to bulk
goethite.

### 3.5.3 Ligand complexation and ligand-promoted dissolution

Wiederhold et al. (2006) showed that at equilibrium Fe isotopic composition was heavier
for the Fe(III)-oxalate complex than free Fe(III) in the solution ($\Delta^{56}$Fe = +0.3‰). Another study
(Dideriksen et al., 2008) found that the formation of the Fe(III)-desferrioxamine B complex in
the solution caused Fe isotopic fractionation, and $\Delta^{56}$Fe was determined to be +0.6‰ at
equilibrium. These experimental results (Wiederhold et al., 2006; Dideriksen et al., 2008) are
consistent with theoretical work which predicted enrichment of heavier isotopes in stronger
bonding environments under equilibrium fractionation (Schauble, 2004; Dideriksen et al., 2008;
Ilina et al., 2013).





732 Brantley et al. (2001) investigated ligand-promoted dissolution of hornblende in oxalate

733 (0.024 mmol/L, pH = 7), and found that dissolved Fe was isotopically lighter than that in bulk

734 hornblende ($\Delta^{56}$Fe = -0.3‰). In the early stage of ligand-promoted dissolution of goethite in

735 oxalate (5 mmol/L, pH = 3), significant Fe isotopic fractionation was observed and $\Delta^{56}$Fe could

736 reach as low as -1.2‰ (Wiederhold et al., 2006); after that, $\delta^{56}$Fe of dissolved Fe increased

737 gradually with reaction time, and eventually was even slightly larger than bulk goethite due to

738 equilibrium isotope effects. Significant Fe isotopic fractionation occurred during ligand-

739 promoted dissolution (5 mmol/L oxalate) of granite and basalt at room temperature (Chapman

740 et al., 2009), and $\Delta^{56}$Fe was down to -1.3‰; similarly, Kiczka et al. (2010) found that ligand-

741 promoted dissolution (5 mmol/L oxalate) of biotite and chlorite resulted in Fe isotopic

742 fractionation ($\Delta^{56}$Fe as low as -0.5‰). In summary, ligand-promoted dissolution resulted in

743 significant Fe isotopic fractionation, and the dissolved Fe appeared to be isotopically lighter

744 due to kinetic fractionation effects.

745 **3.5.4 Discussion**

746 Previous studies, as summarized in Table 1, provide important insights into Fe isotopic

747 fractionation induced by chemical processes which may also occur in the atmosphere. However,

748 experimental conditions used in these studies, including minerals examined, may not be of

749 direct relevance for atmospheric aerosols. Recently, some work started to explore the effects of

750 atmospheric chemical processing on Fe isotopic fractionation, as showcased below.

751 Mulholland et al. (2021) studied Fe isotope fractionation during dissolution of Fe-Mn

752 alloy metallurgy fly ash in synthetic cloud water (pH = 2) under UV/VIS radiation from a solar

753 simulator. Compared to Fe in fly ash, dissolved Fe was isotopically lighter in the first stage of



dissolution (0-60 min) due to kinetic isotopic effects, with $\Delta^{56}$Fe as low as -0.28±0.10‰
(Mulholland et al., 2021); in the second stage of dissolution (60-120 min), dissolved Fe was
isotopically heavier due to equilibrium isotopic effects, with $\Delta^{56}$Fe as high as +0.23±0.09‰
(Mulholland et al., 2021). A following study (Maters et al., 2022) further investigated Fe
isotope fractionation during the initial stage (0-60 min) of dissolution of Tunisian desert dust
and Fe-Mn alloy metallurgy fly ash; compared to Mulholland et al. (2021), the synthetic cloud
water used by Maters et al. (2022) additionally contained 1 mmol/L oxalate. Compared to Fe
in original particles, dissolved Fe was isotopically lighter for both desert dust and fly ash
(Maters et al., 2022), and the extent of Fe isotopic fractionation during dissolution appeared to
be larger for desert dust.
**4. Perspectives**
In the last 10-20 years Fe isotopic analysis has been increasingly used in aerosol research
(as discussed in Section 3), and has been demonstrated to be a promising way to differentiate
sources of total and soluble aerosol Fe, and to quantify the relative importance of different
sources. Several future research directions are proposed here to further enhance the usefulness
of Fe isotopes in atmospheric aerosol research.
(1) The precision of $\delta^{56}$Fe measurements via MC-ICP-MS can reach ±0.02 to ±0.06‰ (2σ)
at present, and this should be enough for most applications in atmospheric aerosols. The
optimal mass of Fe required for isotopic analysis is around 20-100 ng. It is usually not difficult
to collect enough total aerosol Fe for isotopic analysis with acceptable uncertainties, but can
be challenging to collect enough soluble aerosol Fe. Therefore, further improvement in



analytical methods to reduce the minimum mass of Fe required is warranted to increase the
application of Fe isotopic analysis in aerosol research.
(2) Using Fe isotopes to constrain the sources of atmospheric aerosol Fe requires
constraints on isotopic signatures ($\delta^{56}$Fe endmember values) of aerosol Fe from different
sources. The $\delta^{56}$Fe endmember values are reasonably well understood for desert dust Fe, being
rather homogeneous and similar to UCC. However, so far only a limited number of studies
have measured $\delta^{56}$Fe endmember values of non-desert-dust Fe (Table S1), and the number of
samples covered by each of these studies is typically small. The $\delta^{56}$Fe endmember values have
large uncertainties for aerosol Fe from various non-desert-dust sources. Therefore, relevant
experimental measurements are highly needed to reduce these uncertainties.
(3) Up to now only a small number of studies (<20, as summarized in Table 2) have
measured isotopic compositions of ambient aerosol Fe, and the isotopic compositions were less
measured for soluble aerosol Fe than total aerosol Fe. Further application of Fe isotopes in
ambient aerosol research is encouraged, especially in HNLC regions where aerosol Fe
deposition may have large impacts on marine primary productivity and in regions where
anthropogenic and wildfire emission may have substantial contribution to total and soluble
aerosol Fe. Furthermore, additional insights can be gained via comparing source apportionment
results constrained using Fe isotopes with those obtained using correlation (Chuang et al., 2005;
Sholkovitz et al., 2009; Zhang et al., 2023) and factor analysis (Zhu et al., 2020; Chen et al.,
2024; Zhang et al., 2024).
(4) At present, isotopic compositions are assumed to be identical for total and soluble
aerosol Fe from a given source when Fe isotopes are used to constrain the sources of ambient



aerosol Fe. This assumption is yet to be verified as some laboratory studies indicate that
chemical processing in the atmosphere may lead to Fe isotopic fractionation (Section 3.5). As
a result, further studies should be carried out to understand isotopic fractionation of aerosol Fe
caused by chemical processing under atmospherically relevant conditions.

(5) We also encourage modeling studies to include isotopic compositions of aerosol Fe.

Comparison of modelled isotopic compositions of total and soluble aerosol Fe with
measurements (the number of which is increasing) can provide additional constraints to their
sources, in addition to using spatial and temporal variability of their concentrations.

(6) Particle size distribution and mineralogy play a critical role in deposition of soluble

aerosol Fe, because particle size largely dictates sources, chemical processing and fractional
solubility of aerosol Fe (Zhang et al., 2022; Chen et al., 2024) as well as its lifetime, transport
and deposition (Liu et al., 2024). A few studies (Mead et al., 2013; Kurisu et al., 2016a; Kurisu
et al., 2019; Kurisu and Takahashi, 2019; Kurisu et al., 2021; Kurisu et al., 2024) suggest that
isotopic compositions of total and soluble aerosol Fe change significantly with particle size,
implying variation of their sources with particle size. As a result, size-dependence of isotopic
compositions of total and soluble Fe deserves further investigation.

(7) Modeling work (Myriokefalitakis et al., 2015; Hamilton et al., 2020b; Hamilton et al.,

2020a; Bergas-Massó et al., 2023) suggests that when compared to the preindustrial era, the
relative contribution of desert dust, anthropogenic emission and biomass burning to total and
soluble aerosol Fe may have changed immensely. Measurements of isotopic compositions of
total and dissolved Fe in ice core samples (Conway et al., 2015; Xiao et al., 2020) can provide
valuable data to verify historical changes given by modeling simulations.






**Data availability.**

Data used in this review paper all come from previous studies.

**Author contribution.**

**Yifan Zhang:** formal analysis, writing – original draft, writing – review & editing;

**Rui Li:** formal analysis, writing – original draft, writing – review & editing;

**Zachary B. Bunnell:** writing – original draft, writing – review & editing;

**Yizhu Chen:** writing – original draft, writing – review & editing;

**Guanhong Zhu:** writing – review & editing;

**Jinlong Ma:** writing – review & editing;

**Guohua Zhang:** writing – review & editing;

**Tim M. Conway:** conceptualization, writing – original draft, writing – review & editing;

**Mingjin Tang:** conceptualization, formal analysis, writing – original draft, writing – review & editing.

**Competing interests.**

The authors declare that they have no conflict of interest.

**Financial support.**

This work was sponsored by National Natural Science Foundation of China (42321003 and 42277088), Guangzhou Bureau of Science and Technology (2024A04J6533), Guangdong Foundation for Program of Science and Technology Research (2023B1212060049), and Scientific Committee on Oceanic Research (SCOR) Working Group 167 (Reducing



Uncertainty in Soluble aerosol Trace Element Deposition, RUSTED). Zachary B. Bunnell and
Tim M. Conway were supported by NSF Award OCE-1737136.




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
