# Peer review of "A critical review of the use of iron isotopes in atmospheric aerosol research 1 2 Yifan Zhang1,5,#, Rui Li2,#, Zachary B. Bunnell3, Yizhu Chen1, Guanhong Zhu4, Jinlong Ma4, 3 Guohua Zhang, 1 Tim M. Conway3,\*, Mingjin Tang1,6,\* 4 5 1 State Key Laboratory of Advanced Environmental Technology and Guangdong Key 6 7 Laboratory of Environmental Protection and Resources Utilization, Guangzhou Institute"

_EGUsphere, 2025_

## Author Comment (AC1)

Comments by referees are in blue.

Our replies are in black.

Changes to the manuscript are highlighted in red both here and in the revised manuscript.

**Reply to referee #1**

This paper provides a comprehensive summary of iron isotopes in aerosols, including analytical methods. Each paper is carefully read and explained, but I believe further discussion is necessary to integrate the findings more cohesively.

**Reply:** We would like to thank ref#1 for recommending our manuscript for publication and his/her comments which have help us further improve our manuscript. We have addressed these comments and updated the manuscript accordingly; when we do not quite agree with ref#1, we have provided proper explanation. Please find more details below.

For example, a more comprehensive discussion could be achieved by taking into account previous data, such as (i) differences in δ56Fe endmembers among different papers and (ii) the extent to which isotope fractionation caused by chemical processes in the atmosphere actually alters the δ56Fe composition of soluble iron in aerosols.

**Reply:** For studies which measure isotopic composition of aerosol Fe in the atmosphere, the authors normally chose endmember values which can best explain their measurement (typically the lowest $\delta^{56}$Fe they measured in a given study). We are not sure whether and to which extent comparison of endmember values used in different studies can help. This is why we use Section 3.2 to discuss previous studies which reported $\delta^{56}$Fe for aerosol Fe from different sources.

Very few studies which measured $\delta^{56}$Fe in the atmosphere considered isotopic fractionation, and the extent to which isotope fractionation caused by chemical processes in the atmosphere actually alters $\delta^{56}$Fe of soluble aerosol Fe is far from being understood. However, Conway et al (2019) showed no isotopic fractionation of the soluble Fe when looking at aerosols collected near the Sahara. This is a topic which should be considered in future work, as we underscore in the perspective (Section 4). This is also why we use Section 3.5 to discuss previous laboratory studies which examined isotope fractionation. To make it clearer, in the revised manuscript (page 29) we have made the following change to the first paragraph of Section 3.5: "Chemical processes which change the speciation of Fe in the environment may also lead to isotopic fractionation (Table S1), and thereby may attenuate Fe isotopic signatures observed in ambient aerosols. However, when

using Fe isotopic composition to trace sources of total and soluble aerosol Fe, only a few studies have discussed possible Fe isotopic fractionation induced by atmospheric chemical processing (Labatut et al., 2014; Camin et al., 2024); most previous studies assumed that total and soluble Fe from a given source have the same endmember value, implicitly assuming no Fe isotopic fractionation. On the other hand, Conway et al. (2019) showed that total, water-soluble and seawater-soluble Fe in Saharan dust aerosol over the North Atlantic all had the equivalent $\delta^{56}$Fe to the UCC, suggesting no Fe isotopic fractionation and supporting the use of endmembers without fractionation."

Additionally, I suggest reconsidering the structure of the manuscript. For example, some of the discussions in each subsection in Section 3 are only summaries of the other subsections or continuation of explanations; it would be more effective to create a separate discussion section to provide a more comprehensive synthesis.

**Reply:** We respect but do not quite agree with this comment. We have Sections 3.2.4, 3.3.3 and 3.5.5 to summarize Section 3.2, 3.3 and 3.5, respectively. We feel that they are very necessary, because 1) the contents presented in Sections 3.2, 3.3 and 3.5 are very rich; 2) Section 3.2, 3.3 and 3.5 are focused on different aspects. For example, summarizing and discussing what is known for endmembers values (Section 3.2.4) helps us discuss each papers in Section 3.3 (Fe isotopic composition of ambient aerosols).

Also, the same references appear several times in one paragraph, making it difficult to read, and it would be better to reduce such redundancies.

**Reply:** It is a good suggestion. In the revised manuscript we have tried our best to reduce such redundancies if this does not reduce the clarity.

Specific points are described below:

L. 111: It is unclear what "marine source materials" means.

**Reply:** It means potential sources of Fe to the ocean. In the revised manuscript (page 6) we have made the following change to make it clearer: "this parameter has been measured both in potential source materials of Fe to the ocean (Beard et al., 2003a), and later in seawater itself…"

L. 136: Please spell out the "IRMM" at first mention.

**Reply:** As suggested, in the revised manuscript (page 7) we have defined IRMM: "…reported relative to the IRMM-014 standard (provided by the Institute for Reference Materials and Measurements, IRMM)…"

Section 2.2: I suggest adding information on the required Fe amount for TIMS measurement and the purification method for MC-ICP-MS analysis.

**Reply:** To provide information on the required Fe mass for TIMS measurement, in the revised manuscript (page 7) we have made the following change: "In early works, TIMS was combined with the double-spike technique, requiring a minimum Fe mass of >1 µg and yielding precision of ±0.5‰…"

Since Fe isotopic analysis is not a focus of our review paper, we only mention its key aspects and do not discuss details. Instead of providing a detailed description of Fe purification, in the revised manuscript (page 8) we have added one sentence to refer readers to literature: "…samples must be cleanly purified from interfering isobars (Cr and Ni) and matrix elements (e.g., Ca), and further information on Fe purification can be found elsewhere (Conway et al., 2013; Sieber et al., 2021)."

L. 179: Please include the procedural blank value.

**Reply:** In the revised manuscript (page 9) we have included information for the procedural blank: "a concentration that is >20-40 times of our chemistry procedural blank which is typically <0.5 ng/g."

L. 188-191: I recommend moving this paragraph to another subsection within Section 2 and providing further explanation regarding aerosol sample processing, including sampling, acid digestion, extraction procedure, and so on. Otherwise, the explanation seems a bit abrupt.

**Reply:** We agree that in our original manuscript this short paragraph appears to be abrupt, because its connection with previous paragraphs (which are focused on instrumental analysis) is not very clear. As a result, we have made the following change in the revised manuscript (page 9): "In addition to instrumental analysis, a further consideration for aerosol samples that often include significant filter or digestion blanks is the blank isotopic composition…"

Instead of providing further details on aerosol sampling, processing, and so on, in the revised manuscript (page 9) we have added two references (Kurisu et al., 2024; Bunnell et al., 2025) from which interested readers can find more information.

Section 3.2: It would be more effective to summarize previous aerosol δ56Fe results in a figure.

**Reply:** Two previous review articles (Wang et al., 2022; Wei et al., 2024) provide figures similar to what referee #1 suggested. As a result, we choose to provide a comprehensive

compilation of results reported in individual studies (Table S1), instead of using a figure similar to those which can be found in elsewhere.

Table S1 and S2: Please add more information such as: (1) A simple explanation of the sample (e.g. whether it was collected near specific sources or in urban/suburban areas); (2) Soluble Fe extraction method (if available); (3) If it is a certified reference material, the name of CRM.

**Reply:** Here ref#1 has three comments, and they are addressed below.

1) Tables S1 and S2 summarize source materials and ambient aerosols, respectively, and thus we feel the explanation ref#1 suggested is not necessary.

2) It is difficult to add another column into one of the two tables, and therefore we decide not include soluble Fe extraction method in these two tables. Instead, we have include an appendix in the revised manuscript (page 36-37) where we define and discuss total and dissolved Fe (and also several other terms).

3) Usually each individual study included in Table 1 examined more than one sample, and we provide the range and the average for each individual study; as a result, it is not possible for this table to provide the names of CRM used.

In summary, although it would be nice to provide additional information in Tables S1 and S2, we can only provide key information in these tables due to space limit.

L. 259: Biomass burning is not mentioned in Section 3.2.3.

**Reply:** We agree with ref #1, and have made the following change in the revised manuscript (page 12): "…and aerosol particles emitted by various anthropogenic sources (Section 3.2.3)."

L. 262: This sentence could be omitted.

**Reply:** We would like to keep this sentence, as this sentence informs the reader the content of Section 3.2.1 in a concise way.

L. 302~: An explanation of Strategies I and II is necessary here.

**Reply:** It is a good suggestion. In response to this comment, in the revised manuscript we have provided explanation of strategies I and II plants in the appendix (page 36-37).

L. 313~: I found it unclear that Strategy I is isotopically lighter than Strategy II; it rather appears that Strategy II yields lower δ56Fe values. This might be due to δ56Fe being shown as a range. Please consider a more effective way for a clearer explanation. In addition, the reasons for the different δ56Fe values between Strategies I and II should be included.

**Reply:** The referee raised a very good point. Indeed the ranges we provide in the original manuscript seem to suggest Fe in Strategy II plants are lighter. Unfortunately the paper by Kubik et al. (2021) only presents the data graphically, and it is not possible to obtain average or median values. As a result, in the revised manuscript we have removed this sentence (which presents $\delta^{56}$Fe ranges for Strategy I and II plants) to avoid confusion. In the revised manuscript we have added one sentence to explain why $\delta^{56}$Fe are different for strategy I and II plants. We incline to keep this explanation very short and refer interested readers to literature, because our review article is focused on isotopic composition of aerosol Fe (rather than plant Fe).

To summarize, in the revised manuscript (page 15), the discussion of the work by Kubik et al. is given below: "Kubik et al. (2021) surveyed Fe isotopic compositions of biological samples and found that in general Fe in strategy I plants is isotopically lighter than strategy II plants. This is because strategy I and II plants use different biochemical mechanisms to absorb Fe from soil (Kubik et al., 2021)."

L. 324~: This paragraph may be unnecessary.

**Reply:** This paragraph summarizes Section 3.2.3, and more importantly, it points out that Fe isotopic composition is not necessarily identical for source materials and aerosol particles emitted. As a result, we feel that it is necessary and would like to keep it.

L. 330: I found the title "anthropogenic aerosols" unclear; the relationships among Sections 3.2.2, 3.2.3, and 3.3 were not obvious. I understood that 3.2.3 describes materials that can be atmospheric anthropogenic aerosols. Therefore, I suggest changing the title to something like "Materials that can be emitted as aerosol particles" or simply "fly ash and road dust". It would also be beneficial to review the structure of 3.2.2 and 3.2.3 to determine which discussion should be in the same section or not. Combining the two sections into one would be another option.

**Reply:** We agree that the title of Section 3.2.3 in our original manuscript is somehow unclear, and in the revised manuscript (page 15) we have changed it to "Aerosol particles emitted by anthropogenic sources".

It may be an alternative option to combine Sections 3.2.2 and 3.2.3. Nevertheless, we would like to keep them separated because (1) these two sections discuss different contents (although related) and (2) if we combine Sections 3.2.2 and 3.2.3, the section we obtain will be very long.

L. 353~: This paragraph may not be necessary or could be simplified.

**Reply:** This paragraph summarizes what we present in the previous paragraphs and sets the background for the following paragraph; as a result, we think it is necessary. This paragraph only contains two sentences and is not long. As suggested by the referee in previous comment. in the revised manuscript we have deleted unnecessary reference in this paragraph to increase its readability.

L. 380~: As mentioned earlier, this section would be better placed in Section 4 or another discussion section.

**Reply:** Section 3.2.4, which has about 20 lines, summarizes and discusses what we present in Section 3.2, and therefore we would like to keep it at the end of Section 3.2. Indeed a more concise summary (line 777-784, only 8 lines) can be found in Section 4.

L. 406: The meaning of "total" should be clarified, such as by stating "total (acid digested)."

**Reply:** As suggested, we have added one appendix in the revised manuscript (36-37) to define discuss several terms, including total and dissolved Fe.

L. 409: It may be more effective to organize sections based on a different perspective (e.g., sampling location (land or ocean) or particle size separation) rather than publication year.

**Reply:** We have thought about organizing published studied in different ways. It is possible to organize these studies based on sampling location (land or ocean); nevertheless, aerosol Fe does not have marine sources, and it may not help much to organize these studies based on whether sampling was taken over the ocean or land. If we organize these studies based on particle size, it makes comparison of Fe isotopic compositions of different sizes less direct. At the end, we choose to organize them based on publication years.

L. 425: PM should be spelled out.

**Reply:** We agree with the referee. In response to this comment, in the revised manuscript (page 36-37) we have defined $PM_{2.5}$, $PM_{<2.5}$ and $PM_{10}$ in the appendix.

L. 529: The endmember values should be mentioned.

**Reply:** In the revised manuscript (page 23-24), we have made the following change to mention the endmember values Kurisu et al. used: "…the relative contribution of combustion to total aerosol Fe could be up to 50% and 6% for fine and coarse particles, assuming $\delta^{56}Fe$ endmember values to be +0‰ and -4.7‰ to -3.9‰ for dust and combustion Fe."

L. 534: Please clarify whether there is a difference between "fine (<2.5 μm) particles" and PM2.5, and unify the terminology.

**Reply:** Fine and coarse particles are general terms which are vague to some extent. There is no consistent definition of fine and coarse particles, and the cutoff size vary around slightly <1 μm to >2.5μm in different studies. That is why in our manuscript we always try to specify the size range when possible (for example, $PM_{2.5}$, $PM_{10}$, >2.5μm, and etc.).

L. 545: -1.87 and +0.28 should be rounded to -1.9 and +0.3, aligned with "-0.5 to +0.4‰".

**Reply:** As suggested, in the revised manuscript (page 24) we have changed it to "from -1.9 to +0.3‰".

L. 580: As mentioned above, please reconsider the position of this section, and reduce the number of references cited within the same paragraph for better readability.

**Reply:** As suggested, in the revised manuscript we have tried our best to reduce the number of cited references for better readability. Section 3.3.3 summarizes what we present in Sections 3.3.1 and 3.3.2, and therefore we think that it is placed in the right place.

L. 626: Mead et al. (2013), Kurisu et al. (2016), and Li et al. (2022) report δ56Fe values of coal fly ash but do not claim that the heavy δ56Fe in aerosols originates from coal fly ash. Thus, this citation may be inappropriate. It should also be cited that heavy δ56Fe may result from isotopic fractionation in the atmosphere (Labatut et al., 2014; Camin et al., 2024).

**Reply:** Indeed the three studies (Mead et al., 2013; Kurisu et al., 2016; Li et al., 2022) support but do not make the claim that heavy $\delta^{56}Fe$ in aerosols originates from coal fly ash. In the revised manuscript (page 27) we have removed these three references and made the following changes: "This observation has been variably attributed to coal and oil fly ash, to wildfire emissions (Bunnell et al., 2025), or to Fe fractionation due to chemical processing in the atmosphere (Labatut et al., 2014; Camin et al., 2024)."

L. 642: "IMPACT" model should be spelled out.

**Reply:** As suggested, in the revised manuscript (page 28) we have provided the full name of the model: "Kurisu et al. (2021) employed the integrated massively parallel atmospheric chemical transport (IMPACT) model…"

L. 646: Kurisu et al. (2021) suggests the underestimation of mineral dust in coarse particles by the model as well as the inappropriate endmember value of anthropogenic Fe.

**Reply:** We agree with the referee. In fact this has already by discussed in our original manuscript (line 647-650), where it is stated "if the endmember $\delta^{56}Fe$ value of combustion Fe was

set to be larger for coarse particles than fine particles, the model may reproduce the observed $\delta^{56}Fe$ for both fine and coarse particles."

L. 692: Please verify whether the fractionation factor cited here is appropriate. Skulan et al. (2002) suggest the kinetic isotope fractionation factor of hematite precipitation as +1.32‰, meaning the dissolved phase becomes heavier. They also suggest that there is no clear fractionation during the dissolution experiment with 0.9 mol/L HCl.

**Reply:** As suggested, we checked the paper by Skulan et al. (2002) carefully, and agree with the referee that our initial interpretation was incorrect. In the revised manuscript (page 30), we have made the following change: "For hematite dissolution in 0.9 mol/L HCl, no significant equilibrium fractionation of Fe was observed (Skulan et al., 2002)."

L. 700: This part may be unnecessary. Since the conditions of each experiment are different in terms of mineralogy, temperature, solvent, equilibrium/kinetic, etc., it is understandable that the degree of fractionation varies.

**Reply:** We agree with the referee that this paragraph is not very necessary. As a result, we have deleted it in the revised manuscript.

L. 745: The title of this section should be revised, as it is not a discussion in response to the previous section, but simply describes an experiment under real atmospheric conditions.

**Reply:** We agree with the referee. In the revised manuscript (page 32) we have changed the title of Section 3.5.4 to: "Fe isotopic fractionation caused by atmospheric chemical processing".

L. 755: Consider explaining the dissolution rate (%) rather than dissolution time, as it would facilitate comparisons with other studies.

**Reply:** As some other studies did not report dissolution fractions, it is difficult to compare different studies based on dissolution fractions. However, what ref #1 suggested is very good, and in the revised manuscript (page 35) we have added one sentence to recommend this for future work: "We suggest that dissolution fractions (in %) should also be reported together with isotopic fractionation, in order to facilitate comparisons between different studies."

L. 784: Could you clarify what kind of experimental measurements you expect? I believe this is an important perspective that should be addressed.

**Reply:** As suggested, in the revised manuscript (page 34) we have modified this sentence to make our recommendation more explicit: "Therefore, measurements of size-dependent $\delta^{56}Fe$ of

total and soluble Fe in aerosol particles emitted by various anthropogenic and combustion sources are highly recommended."

L. 798-800: Based on the discussion in Section 3.5, if isotopic fractionation of Fe aerosols in the atmosphere is driven by chemical processes, a relationship between solubility and isotopic fractionation (difference between total and soluble) should exist ─ i.e., lower solubility corresponds to larger isotope fractionation. I am curious whether such a relationship can be identified from the compiled data. This discussion should address whether the low $\delta56Fe$ in soluble Fe is due to different sources or chemical processes.

**Reply:** We think that both sources and chemical processes will influence isotopic composition of total and soluble Fe. However, their relative extents are totally unclear ambient aerosol Fe, since most previous studies on aerosol Fe isotopes did not consider Fe isotopic fractionation. This is exactly why we use Section 3.5 to discussion Fe isotopic fractionation induced by chemical processing (we also discuss this issue in Section 4). We hope that this will stimulate future studies to take into account the effects of Fe isotopic fractionation on isotopic composition of aerosol Fe in the atmosphere.

---

## Author Comment (AC2)

Comments by referees are in blue.

Our replies are in black.

Changes to the manuscript are highlighted in red both here and in the revised manuscript.

**Reply to referee #2**

This review is amazingly thorough and much needed by the community as a summary of Fe isotope values as they relate to aerosols. The supplementary tables are invaluable as comprehensive summaries of published Fe isotope values. I am very excited to see this paper published.

**Reply:** We would like to thank ref#2 for recommending our manuscript for publication and his/her comments which have help us further improve our manuscript. We have addressed these comments and updated the manuscript accordingly; when we do not quite agree with ref #2, we have provided proper explanation. Please find more details below.

There is some work, however, that needs to be done to make the manuscript a more cohesive unit. Terms need to be summarized the first time they are used (this is especially true for operational definitions such as "soluble"), and word choice needs to be used more consistently (for example, desert dust and UCC should not be used interchangeable/have different definitions in different sections).

**Reply:** It is a good suggestion. We have made the following changes in the revised manuscript: 1) we have tried to use replace desert dust with UCC whenever possible; 2) for terms which need definition, we have defined them where they appear for the first time or in the appendix (page 36-37).

Further, I suggest re-organizing the manuscript as right now it reads as separately written sections that are sometimes redundant and do not lead the reader to the comprehensive takeaways for different processes.

Right now, I feel like this paper is a very detailed review of all applicable literature, but I think it could be made stronger by streamlining the details and adding an introductory/summary as things relate to aerosols for all sections. This will allow the relevance and summary of the literature to be clear to readers.

Possible new, simplified outline (would require reorganizing current paragraphs into the correct section:

**Reply:** We fully understand that these comments raised ref #2 kindly are intended to help us further improve our manuscript. The structure which ref #2 suggested may also be a good option. However, we prefer to not to change the structure of our manuscript, because in our opinion the current structure delivers very well the information we want to deliver.

We feel that overall our manuscript has been streamlined quite well. For example, we have used Section 1 as the introduction and Section 4 as the summary; in addition, we have also provide necessary introductions or summaries for some key subsections (for example, Sections 3.2 and 3.3).

In the revised manuscript (page 29), we have added a few sentence to inform the readers why we discuss Fe isotopic fractionation in Section 3.5: "Chemical processes which change the speciation of Fe in the environment may also lead to isotopic fractionation (Table S1), and thereby may attenuate Fe isotopic signatures observed in ambient aerosols. However, when using Fe isotopic composition to trace sources of total and soluble aerosol Fe, only a few studies have discussed possible Fe isotopic fractionation induced by atmospheric chemical processing (Labatut et al., 2014; Camin et al., 2024); most previous studies assumed that total and soluble Fe from a given source have the same endmember value, implicitly assuming no Fe isotopic fractionation. On the other hand, Conway et al. (2019) showed that total, water-soluble and seawater-soluble Fe in Saharan dust aerosol over the North Atlantic all had the equivalent $\delta^{56}$Fe to the UCC, suggesting no Fe isotopic fractionation and supporting the use of endmembers without fractionation."

In addition, in the revised manuscript we have also tried to reduce redundancies or to provide further discussion when possible. Please refer to our reply to the following comments and our revised manuscript for more details.

Specific line comments/questions:

L23-40: Abstract would need to be reorganized if the suggestion of changed manuscript outline is selected.

**Reply:** As we prefer not to change the overall structure of our manuscript (please see our reply to previous comments), we decide not to re-organize the abstract either.

L45: Define dissolved, also consider using the abbreviation dFe throughout the paper.

**Reply:** As suggested by both referees, we have added an appendix in the revised manuscript (page 36-37) to define several terms used in our manuscript, including total Fe and dissolved Fe. Since dissolved Fe is not much longer than dFe, we prefer to use dissolved Fe for better readability.

L48-49: Define HNLC

**Reply:** In fact we have already defined HNLC in our original manuscript.

L50: Define the multiple ways you use the word soluble in this paper, i.e. size fraction and chemical leaching method

**Reply:** As suggested by both referees, we have added an appendix in the revised manuscript (page 36-3) to define several terms used in our manuscript, including total Fe and dissolved Fe (soluble Fe). When we define dissolved Fe (soluble Fe), we mention that multiple ways are used in different studies.

L73: The , after troposphere makes it seem like it's part of the list that follows.

**Reply:** To avoid this misunderstanding, in the revised manuscript (page 4) we have changed this sentence to "Natural, anthropogenic and wildfire aerosols may undergo various chemical and physical processes in the atmosphere, which can "solubilize" insoluble Fe minerals to soluble Fe"

L82: At this point, it is difficult to understand what you mean by 'ambient aerosols'. I would explicitly define it. This is an example also of where using desert dust and UCC consistently and making the definitions clear is important.

**Reply:** Here ambient aerosols mean aerosol particles in the troposphere. To explicitly define ambient aerosol, in the revised manuscript (page 4) we have made the following change: "…Fe solubility can be much higher for ambient aerosols (aerosol particles in the troposphere) collected over the oceans…"

We have checked the entire manuscript, and replaced desert dust with UCC when possible.

L95: Define enrichment factors and add a sentence for how they are used before you mention that there isn't a consensus.

**Reply:** As suggested, in the revised manuscript (page 36-37) we have added an appendix where we have defined several terms, including enrichment factors.

L104: Single particle analysis through what methods?

**Reply:** In the revised manuscript (page 6) we have made the following change to mention the typical methods used for single particle analysis: "Single particle analysis, typically using electron microscopy, X-ray micro-spectroscopy or single particle mass spectrometry, can also be very useful for source identification of aerosol Fe in individual particles…" In addition, references cited here have also been updated.

L116-121: An example of a section that can be moved to the modeling section. Also, it was unclear, how they determined coal combustion as a major anthropogenic source. Seems like a key takeaway.

**Reply:** The two papers (Wang et al., 2022; Wei et al., 2024) are review papers which review studies related to aerosol Fe isotopes, and they both use a statistical model (MIXSIAR) to re-analyze literature data. As a result, we mention them in Section 1 (Introduction) instead of in Section 3.4 (modeling studies). In the revised manuscript (page 28), we have also changed the title of Section 3.4 to reflect its content more properly: "3.4 Atmospheric modeling studies of isotopic compositions of ambient aerosol Fe".

It is beyond the scope of our review paper to explain why Wei et al. (2024) identified coal combustion as the major anthropogenic source of aerosol Fe. This is perhaps linked to the endmember values they used for coal combustion.

L137-139: Add information about how IRMM-014 is running out and what standard the field is moving to.

**Reply:** In the revised manuscript (page 7) we have added the following sentence to provide more information related to IRMM-014 and its substitutes: "Production of IRMM-014 has ceased and current stocks will likely run out in the future; as a result, new reference materials, such as IRMM-524a, are beginning to be used in place of IRMM-014 (González De Vega et al., 2020; Xu et al., 2022)."

L148: I would argue that 1-3permille is a more realistic range across transects.

**Reply:** We agree with the referee, and have changed it to "…typically 1-3‰…" in the revised manuscript (page 8).

L150: short -> shorter

**Reply:** In the revised manuscript (page 8) we have implemented the suggested change.

L172: Choose either SE, 1sigma, and 2sigma error for consistent notation throughout the paper.

**Reply:** In our manuscript we usually use $2\sigma$ to represent errors. In response to this specific comment, in the revised manuscript (page 8) we have made the following change: "…to achieve acceptable uncertainties (<0.15‰, $2\sigma$) in solutions…"

As suggested, in the revised manuscript we have made changes so that we always used.

L174-187: Talking in ng of Fe is not common for many of the target readers of this article. I would consider explaining how you get these numbers from concentration to make sure readers are on the same page.

**Reply:** The minimum amount of Fe required by Fe isotope analysis, is typically expressed in mass. Interested readers can easily obtain the minimum concentration of aerosol Fe (in the troposphere) required if they know the air volume samples (and vice versa).

L174: Add "of Fe" after a minimum mass

**Reply:** We have made the suggested change in the revised manuscript (page 8): "…a minimum mass of Fe depends on…"

L186: Define "such precision"

**Reply:** As suggested, in the revised manuscript (page 8) we have made the following change to define the precision: "Such precision (<0.1‰) is often essential for resolving variability between aerosol samples."

L191: Add sources

**Reply:** We have carefully checked L191. We feel that we cannot add "sources" to this line and that the sentence to which L191 belongs looks fine.

L198: This is an example of a section that is out of place in my opinion. It comes right after you finish explaining that you won't discuss in-depth marine studies. It also references fractionation processes in the surface that you have yet to discuss in the manuscript. Also, it is unclear how you call things "dust" when you later explicitly state that the surface has to be ignored through fractionation.

**Reply:** We feel that Section 3.1 is in the right place. Before we review application of Fe isotopic analysis in aerosol research, we have a short section (Section 3.1) we present a few examples of relevant ocean observational and modeling studies to illustrate how $\delta^{56}$Fe may help constrain sources of dissolved Fe to the ocean, focusing on aerosol deposition case studies; in other words, Section 3.1 sets the context for the following contents in Section 2. This is mentioned at the beginning of Section 3 and further elaborated in the first paragraph of Section 3.1.

We mention fractionation when we discuss the three examples used in Section 3.1, because the measured Fe isotopic composition of a sample is not only affected by its sources, but may also be impacted by isotopic fractionation.

L203: Different precision range than earlier in the manuscript.

**Reply:** To keep the precision ranges consistent, in the revised manuscript (page 10) we have made the following change: "…with a precision of better than ±0.10‰."

L220: Confusing that you call it 2 component mixing model when there are more than two sources.

**Reply:** At each station, Fe is assumed to originate from two sources, and one source is always dust, and the other source varies with stations. In the revised manuscript (page 11), we have made the following change for better clarification: "…two component mixing was used to quantitatively constrain sources at each station across the basin…"

Because 1) we only use this study (Conway and John, 2014) as one of the three examples to demonstrate the usefulness of Fe isotopes to understand the sources and processes of marine Fe and 2) the paragraph is already quite long, we decide not to provide further details but refer the interested readers to the work by Conway and John (2014).

L226: I recommend adding more sources that discuss the fractionation process at the surface ocean.

**Reply:** Similar to our reply to last comment, we only use this study (Conway and John, 2014) as one of the three examples to demonstrate the usefulness of Fe isotopes to understand the sources and processes of marine Fe, and it is beyond the scope of our manuscript to discuss in depth Fe isotope fractionation at the surface ocean; in addition, the paragraph is already quite long. As a result, we prefer not to discuss the fractionation process at the surface ocean in more details.

L247: This would probably make more sense being moved to the modeling section.

**Reply:** We respectively disagree with the referee. The modeling study by Konig et al. (2021) is focused on biogeochemical processes of Fe in the ocean, and we use this study as one of the three examples to demonstrate the usefulness of Fe isotopes to understand the sources and processes of marine Fe; on the other hand, Section 3.4 only discusses modeling studies of aerosol Fe in the atmosphere. As a result, we prefer to discuss the work by Konig et al. (2021) in Section 3.1.

To better reflect the content of Section 3.4, in the revised manuscript (page 34), we have changed its title: "3.4 Atmospheric modeling studies of isotopic compositions of ambient aerosol Fe".

L256: More background information on how aerosols are shown to be a mixture of natural and anthropogenic sources and choose the words that fit into each category that you will consistently use throughout.

**Reply:** We have a paragraph in our original manuscript (line 59-72, Section 1) to discuss natural and anthropogenic sources of aerosol Fe in the atmosphere; as a result, we prefer not to repeat it in Section 3.2.

We agree with the suggestion on using consistent terms for aerosol sources. In the revised manuscript, we have made the following changes (page 12): "…and aerosol particles emitted by various anthropogenic sources...", and updated the title of Section 3.2.3 (page 15): "3.2.3 Aerosol particles emitted by anthropogenic sources".

L275-279: Missing + before heavier Fe isotope values.

**Reply:** We have checked the entire manuscript (and supplement), and added "+" for all the positive $\delta^{56}$Fe values.

L276: Define ATD.

**Reply:** As suggested, in the revised manuscript (page 13) we have defined ATD: "…namely Arizona Test Dust (ATD)…"

L288: Calculate an actual average or state something like, "for natural dust contributions, we can assume an endmember equal to that of UCC (+0.09 permille). Also, I think it is important to mention in the natural aerosol section that not all natural aerosols are from deserts.

**Reply:** As suggested, in the revised manuscript (page 14) we have modified the first sentence in this paragraph: "Most of the $\delta^{56}$Fe values for desert dust Fe, as reported in previous studies, fall into a small range (-0.1 to +0.19‰); as a result, one may conclude that the Fe isotopic composition of desert dust is very similar to the UCC, characterized by an endmember value of around +0.09‰."

In addition, we have also added one sentence in the revised manuscript (page 14) to mention that not all the natural aerosol Fe originates from desert dust: "In addition to desert dust, there are other natural sources for aerosol Fe which may have different isotopic composition, such as soil particles entrained into the atmosphere during wildfires (Hamilton et al., 2022)."

L353-261: This is a paragraph that I think would be a better introduction to the anthropogenic aerosol section.

**Reply:** This paragraph serves to summarize the first three paragraphs in Section 3.2.3, and it also sets the background for next paragraph (also the last paragraph in Section 3.2.3). As a result, we believe that it is suitable to keep it in the current place.

L368: If you are using leach data, talk about how the leaches are proven to fractionate or not fractionate Fe and the sources that support this.

**Reply:** Most of studies reviewed in Sections 3.2 and 3.3 does not consider Fe isotopic fractionation, and this is why we discuss Fe isotopic fractionation in Section 3.5. This has been discussed in our original manuscript (page 38, line 795-797).

To make this more explicit, we have made the following change in the revised manuscript (page 29): "Chemical processes which change the speciation of Fe in the environment may also lead to isotopic fractionation (Table S1), and thereby may attenuate Fe isotopic signatures observed in ambient aerosols. However, when using Fe isotopic composition to trace sources of total and soluble aerosol Fe, only a few studies have discussed possible Fe isotopic fractionation induced by atmospheric chemical processing (Labatut et al., 2014; Camin et al., 2024); most previous studies assumed that total and soluble Fe from a given source have the same endmember value, implicitly assuming no Fe isotopic fractionation. On the other hand, Conway et al. (2019) showed that total, water-soluble and seawater-soluble Fe in Saharan dust aerosol over the North Atlantic all had the equivalent $\delta^{56}$Fe to the UCC, suggesting no Fe isotopic fractionation and supporting the use of endmembers without fractionation."

L381-391: I think this paragraph would be a better introduction to an endmembers section.

**Reply:** This paragraph and the other paragraph in Section 3.2.4, are used to summarize previous studies on endmember values. As a result, instead of putting it in the front of Section 3.2 (as an introduction), we prefer to have it as the end of Section 3.2 (as a summary).

L399: Define "fine" and "coarse" and let the reader know if these definitions change from study to study.

**Reply:** The cutoff size between fine and coarse particles varies in different studies. That is why we provide the size ranges for fine and coarse particles in our manuscript whenever possible.

L409: If you keep this outline, why is there a 2016 cutoff?

**Reply:** In the revised manuscript (page 18) we have added one sentence to explain why we choose 2016 as the cutoff year: "We choose the year of 2016 as the cutoff, because Kurisu and co-workers who published several papers in this field published the first paper in 2016."

L411: Define TSP

**Reply:** We have already defined TSP previously (line 367 in our original manuscript).

L458: These are the lightest isotope values included in the paper. Might be worth positing why it makes sense they are so light (or why it is surprising).

**Reply:** In fact the lightest Fe so far was reported by Hsieh and Ho (2024). It is a good idea to highlight the lightest Fe. In the revised manuscript (page 25) we have added one sentence to highlight it: "This study reported lowest $\delta^{56}$Fe values so far for total and soluble Fe in the

troposphere (down to -3.35 and -4.46‰, see Table S2)." The possible reason for the highest Fe is implied in the sentence immediately after the sentence we have added.

L480-501: This is an example location where I think this information would flow better in the applicable endmember section.

**Reply:** The work by Kurisu and Takahashi (2019) investigated ambient aerosol impacted by biomass burning, and can also be placed in Section 3.2.3 where we discuss aerosol particles emitted by anthropogenic sources. However, because the ambient aerosols examined by Kurisu and Takahashi (2019) also originated from other sources (in addition to biomass burning), we prefer to discuss this work in Section 3.3 where we discuss Fe isotopic composition of ambient aerosols.

L519-579: These sections are examples of where it feels like there was a different author. They are a lot more detailed/lend a full paragraph to one study, compared to elsewhere in the manuscript. I think it would help the reader to streamline this and draw connections between the studies as applicable.

**Reply:** The two papers (Kurisu et al., 2021, Bunnell et al., 2025) are key studies and present very rich and insightful information; therefore, the two paragraphs which analyze and discuss these two studies are quite long. Understandably, different people have different opinions on the extent to which details should be provided; for example, ref #1 asked us to provide endmember values Kurisu et al. used in their two component mixing model.

Nevertheless, we agree with ref #2 in principle, and have tried our best to make these two paragraphs shorter. The paragraph which discusses the work by Kurisu et al. (2021) has been shortened from 15 to 13 lines (page 23-24 in the revised manuscript), and the one discusses the work by Bunnell et al. (2025) has been reduced from 17 to 16 lines (page 25 in the revised manuscript).

L644: Discuss why different endmembers were used in this study.

**Reply:** We carefully checked this paper (Kurisu et al., 2021) which states "The representative $\delta^{56}$Fe value of combustion Fe was estimated to be -3.9‰ to -4.7‰ based on the $\delta^{56}$Fe values of the aerosol samples collected at various sites, which included suburban areas and sites near sources of anthropogenic emissions." We know how the authors assigned endmember values to combustion Fe, but do not know why they used different values; as a result, we are not able to discuss why different endmember values were used by Kurisu et al. (2021).

L644: I understand why you have included this section, as aerosols are greatly chemically altered before deposition, however this has not been explained to the readers. If you leave these sections, make sure to include examples of how this is applicable to aerosol studies and the main takeaways aerosol people should know from this information.

**Reply:** This is a good suggestion. In the revised manuscript (page 29) we have made the following change to inform the readers why we discuss Fe isotopic fractionation in Section 3.5: "Chemical processes which change the speciation of Fe in the environment may also lead to isotopic fractionation (Table S1), and thereby may attenuate Fe isotopic signatures observed in ambient aerosols. However, when using Fe isotopic composition to trace sources of total and soluble aerosol Fe, only a few studies have discussed possible Fe isotopic fractionation induced by atmospheric chemical processing (Labatut et al., 2014; Camin et al., 2024); most previous studies assumed that total and soluble Fe from a given source have the same endmember value, implicitly assuming no Fe isotopic fractionation. On the other hand, Conway et al. (2019) showed that total, water-soluble and seawater-soluble Fe in Saharan dust aerosol over the North Atlantic all had the equivalent $\delta^{56}$Fe to the UCC, suggesting no Fe isotopic fractionation and supporting the use of endmembers without fractionation."

L764: I really liked the perspectives suggestion, I would just suggest that you make it even more strongly stated that these are the recommended steps that Fe isotope aerosol research needs to take. You say it, but stronger language would make it stand out. Also make sure to have some sort of conclusion at the end and that the manuscript doesn't just end at the end of the last suggestion.

**Reply:** We would like to thank referee #2 for his/her highly positive comment on Section 4. As suggested, we have made the following two changes to make our recommendation stronger. The first change is (page 33): "After reviewing and discussing previous studies in a critical manner, we recommend several future research directions below to further enhance the usefulness of Fe isotopes in atmospheric aerosol research", and the second change is (page 34): "Therefore, measurements of size-dependent $\delta^{56}$Fe of total and soluble Fe in aerosol particles emitted by various anthropogenic and combustion sources are highly recommended."

We think we have already have a number of conclusions in Section 4 which has eight paragraphs. In each of the last seven paragraphs, we first provide a summary/conclusion in one aspect, and discuss future work we recommend in this direction.

Fig 1: Not everything in this figure is discussed in the paper (i.e. SO2, NOx, VOC, DMS, CO2). Either include them in the fractionation process section or remove them from the figure.

**Reply:** We use Figure 1 to provide a schematic overview of emission, transport, processing and deposition of aerosol Fe, and thus I put in in Section 1. We include trace gases (such as $SO_2$ and $NO_x$) in this figure to illustrate that reactions of these trace gases lead to aging of aerosol particles (including aerosol Fe).

Fig 4: I think this figure would be even stronger if there was some indication on the figure itself of the concentration differences between the two seasons.

**Reply:** This figure comes from a previous paper by Kurisu et al., and it is difficult for us to add another information to this figure. Therefore, we prefer to use this figure as it is. Please note that we have underscored and discussed the differences in Fe solubility in our original manuscript (line 463-466).

Supp Table S1&S2: Make sure to define your size fractions

**Reply:** In our original manuscript, we have provided size information in Tables S1 and S2 (ambient aerosols) whenever possible. Please note that the results summarized in Table S2 usually do not contain size information (in many cases the samples examined are not aerosol particles).

---

## Author Comment (AC3)

Comments by Capucine Camin are in blue.

Our replies are in black.

Changes to the manuscript are highlighted in red both here and in the revised manuscript.

**Reply to Capucine Camin**

Great manuscript! I would like to inform you about this work: https://doi.org/10.5194/egusphere-2024-3777 (the latest version is in the discussion section).

**Reply:** We would like to thank Camin et al. for bringing their recent manuscript to our attention. We have made the following changes in the revised manuscript accordingly. We have added one paragraph to this work at the end of Section 3.3.2 (page 26), and mentioned it at the end of Section 3.3.3 (page 27) and at the beginning of Section 3.5 (page 29).